# Designed 2D protein crystals as dynamic molecular gatekeepers for a solid-state device

Sanahan Vijayakumar[1,7], Robert G. Alberstein[2,7], Zhiyin Zhang[2,7], Yi-Sheng Lu[1], Adriano Chan[3], Charlotte E. Wahl[4], James S. Ha[4,6], Deborah E. Hunka [4], Gerry R. Boss[3], Michael J. Sailor [1,2,5] ✉ & F. Akif Tezcan [1,2] ✉

The sensitivity and responsiveness of living cells to environmental changes are enabled by dynamic protein structures, inspiring efforts to construct artificial supramolecular protein assemblies. However, despite their sophisticated structures, designed protein assemblies have yet to be incorporated into macroscale devices for real-life applications. We report a 2D crystalline protein assembly of $^{C98/E57/E66}$L-rhamnulose-1-phosphate aldolase ($^{CEE}$RhuA) that selectively blocks or passes molecular species when exposed to a chemical trigger. $^{CEE}$RhuA crystals are engineered via cobalt(II) coordination bonds to undergo a coherent conformational change from a closed state (pore dimensions <1 nm) to an ajar state (pore dimensions ~4 nm) when exposed to an HCN(g) trigger. When layered onto a mesoporous silicon (pSi) photonic crystal optical sensor configured to detect $HCN_{(g)}$, the 2D $^{CEE}$RhuA crystal layer effectively blocks interferents that would otherwise result in a false positive signal. The 2D $^{CEE}$RhuA crystal layer opens in selective response to low-ppm levels of $HCN_{(g)}$, allowing analyte penetration into the pSi sensor layer for detection. These findings illustrate that designed protein assemblies can function as dynamic components of solid-state devices in non-aqueous environments.

Since the isolation of graphene in 2004[1], there has been a tremendous interest in the synthesis of two-dimensional (2D) materials (e.g., boron-nitrides, metal-oxides/hydroxides/carbides/chalcogenides, metal- and covalent-organic frameworks) and the exploitation of their unique electronic properties, planar structures, and high surface-to-volume ratios for applications in catalysis, sensing, and separations[2–11]. The ultimate benchmark for functional 2D materials are biological membranes, composites of lipid bilayers and embedded proteins that define the boundaries of cells[12,13]. Biological membranes regulate the highly selective transport of ions or molecular species in and out of cells, enabling cells to sense their environment, transmit signals, and

perform catalytic reactions linked to energy transformation[12,13]. A major distinction of biological membranes from the aforementioned synthetic 2D materials is their dynamic nature, whereby the natural membrane's selective permeability for ions or molecules is controlled in specific response to environmental stimuli (e.g., pressure, temperature, voltage, ions/molecules)[14–19]. Underlying this responsiveness is the ability of membrane-embedded proteins to undergo specific conformational changes upon exposure to stimuli[14–19]. Emulating these dynamic and responsive properties in synthetic materials is highly desirable, particularly from the standpoint of sensing and separation applications. However, most synthetic 2D materials are structurally

[1]Materials Science and Engineering Program, University of California, San Diego, La Jolla, CA 92093, USA. [2]Department of Chemistry & Biochemistry, University of California, San Diego, La Jolla, CA 92093, USA. [3]Department of Medicine, University of California, San Diego, La Jolla, CA 92093, USA. [4]Leidos, 4161 Campus Point Ct, San Diego, CA 92121, USA. [5]Department of Nanoengineering, University of California, San Diego, La Jolla, CA 92093, USA. [6]Present address: Battelle, 505 King Ave Columbus, Ohio, OH 43201, USA. [7]These authors contributed equally: Sanahan Vijayakumar, Robert G. Alberstein, Zhiyin Zhang. ✉e-mail: msailor@ucsd.edu; tezcan@ucsd.edu

invariant and cannot undergo large structural changes in a stimulus-dependent fashion. Although dynamic/flexible metal- and covalent- 2D frameworks have been developed[20–25], their use in separations applications has relied almost entirely on a passive size-exclusion effect achieved through their multilayered arrangements in membranes[26–28], rather than an active gating mechanism induced by analyte-specific transformations within their 2D structures.

Given their immense structural diversity as well as their chemical and genetic malleability, proteins are attractive building blocks for engineering functional materials with well-defined architectures. Fueled by the development of innovative design strategies in the last decade, researchers have constructed a diverse array of artificial protein assemblies and protein-based materials with well-defined structures[29–34], including crystalline 2D frameworks[35–37]. With increasing frequency, these protein assemblies display dynamic and responsive structures as well as emergent functions[38–41], which have even begun to be integrated into living systems[37,42]. However, such designed protein assemblies have yet to be incorporated into devices whereby their dynamic and responsive properties can be exploited in practical applications. In contrast to the inorganic materials mentioned above, biologic materials are generally less stable and more difficult to incorporate into solid-state devices, especially if the environment of the intended application lies outside the very narrow window of solution conditions from which the biologic was derived. Key questions in this regard are (a) how to physically interface a structurally well-ordered protein-based material with a solid-state device and (b) whether the structural, dynamic, and functional properties of such a protein-based material can be maintained under non-biological conditions encountered during device operation.

This study addresses these questions by integrating a dynamic, 2D protein crystalline material self-assembled from the C98/E57/E66 variant of L-rhamnulose-1-phosphate aldolase ($^{CEE}$RhuA) recently engineered in our laboratories[40] as an analyte-triggerable membrane into a porous silicon (pSi)-based, remote sensing chip. We show that the 2D $^{CEE}$RhuA crystals can be efficiently deposited onto the mesoporous silicon substrate and engineered to trigger open in the presence of hydrogen cyanide (HCN) vapor, a potent asphyxiant and designated as both a chemical warfare agent and a toxic industrial chemical[43]. While many sensors exist for cyanide detection (Supplementary Table 1), we surmised that this pSi modality would provide an ideal platform for determining whether a protein gatekeeper could function in a controllable manner outside of its native aqueous environment, as the flat substrate is geometrically compatible with 2D crystal deposition and its established sensitivity to analyte infiltration provided a clear read-out of the porosity of the protein layers. We show that the gatekeeper rejects chemical interferents that would normally generate substantial zero-point drift in the response from the pSi sensor. Extensive structural, analytical, and computational investigations show that the gatekeeping functionality of $^{CEE}$RhuA crystals is enabled by their ability to undergo coherent changes in permeability in an HCN-specific manner despite being adsorbed onto a solid-state surface and being subjected to environmental conditions that typically degrade the function of isolated proteins.

## Results and discussion
### Design of the 2D protein crystal "gatekeeper"
The construction of the protein gatekeeper-sensor system is illustrated in Fig. 1. The 2D protein crystals to be used as analyte-triggerable membranes were built from L-rhamnulose-1-phosphate aldolase (RhuA)[44], a square-shaped $C_4$-symmetric homotetrameric protein (Fig. 1a). In earlier work, we showed that $^{C98}$RhuA, a variant with Cys residues engineered in its four identical corners (positions 98) assembles via disulfide bond formation into porous, defect-free, and mono-layered 2D crystals with μm-scale dimensions under mildly oxidizing conditions[36]. Remarkably, the C98-C98 disulfide linkages

acted as flexible hinges, enabling $^{C98}$RhuA crystals to undergo a coherent structural phase change that resulted in the opening and closing of interstitial voids between the protein units; these pores could be induced to open to as large as 7 nm and then close to <1 nm without loss of crystallinity. The fully closed state of $^{C98}$RhuA was thermodynamically favored due to solvent entropy effects. In subsequent work[40], the $^{C98}$RhuA crystals were engineered with pairs of negatively charged glutamate residues flanking each corner (E57 and E66), such that the fully closed state of the resulting variant ($^{CEE}$RhuA) is disfavored at thermodynamic equilibrium due to electrostatic repulsion between the glutamates, yielding an "ajar" state with a pore size of ~4 nm (Fig. 1a)[40].

We had previously shown that addition of $Ca^{2+}$ ions induced these 2D crystals of $^{CEE}$RhuA to collapse to their closed form via specific $Ca^{2+}$ coordination to the pairs of E57 and E66 glutamate residues near the hinges while also mitigating their electrostatic repulsion[40]. For the present work, we induced the closed state of $^{CEE}$RhuA using $Co^{2+}$ as a divalent metal ion (Fig. 1b)[40]. We chose $Co^{2+}$ for these experiments because, like $Ca^{2+}$, $Co^{2+}$ exhibits a high affinity for carboxylate-based ligands such as the E57 and E66 residues to effect the closing of the $^{CEE}$RhuA crystals. However, in contrast to $Ca^{2+}$, $Co^{2+}$ forms a very stable hexacyanide complex $[Co(CN)_6]^{4-}$ that we hypothesized would provide a strong and specific driving force to open the gates in the crystal[45]. To verify that $Co^{2+}$ could induce the closed state, clarified suspensions of $^{CEE}$RhuA crystals were mixed in a 1:1 v:v ratio with MES-buffered 50 mM $CoCl_2$ solution and incubated for ≥3 d to allow complete metal binding. The negative-stain transmission electron microscope (ns-TEM) images before and after $Co^{2+}$ treatment indicated that $Co^{2+}$ binding indeed prompted closure of the $^{CEE}$RhuA crystal pores en masse (Fig. 1b). The closure is further evidenced by the reduction of the unit cell dimension $a$ from ~11.4 to ~11.0 nm, as determined from the fast Fourier transforms of the ns-TEM images (Fig. 1b).

We then established the capability of the $^{CEE}$RhuA crystals to be switched from the closed to the ajar state in a cyanide ($CN^-$)-dependent manner in aqueous solution. A mixture of $Co^{2+}$-bound $^{CEE}$RhuA crystal suspensions 1:1 (v:v) with a buffered 150 mM KCN solution (final $CN^-$:$Co^{2+}$:$^{CEE}$RhuA molar ratio was 6:1:0.0002 to fully sequester all $Co^{2+}$ ions, both solvated and gatekeeper-bound) led to the coherent opening of the crystals and restoration of the original unit cell dimension (Fig. 1b). Vigorous mixing of the $Co^{2+}$-bound $^{CEE}$RhuA crystal suspensions, or dialysis to remove free $Co^{2+}$ ions, also resulted in the opening of $^{CEE}$RhuA crystals, consistent with the role of $Co^{2+}$ as a bridging ion. Taken together, these observations demonstrate that not only is the "closed" conformation of a 2D $^{CEE}$RhuA crystal stabilized by $Co^{2+}$ ions, but also that $Co^{2+}$-binding can be reversed upon exposure to $CN^-$, restoring the "ajar" state of the crystal. Thus, the $^{CEE}$RhuA lattice represents a 2D framework possessing molecular-scale gaps that can undergo substantial dimensional changes in specific response to $CN^-$. Importantly, the conformational dynamics of $^{CEE}$RhuA lattices are coherent in that their pore sizes change uniformly and synchronously, which distinguishes them from soft, polymer-based porous materials.

### Construction of the porous Si photonic crystal sensor with a protein crystal gatekeeper
We next deposited the gatekeeper onto the surface of the solid-state sensor (Fig. 1c). The HCN vapor sensing element was a two-dimensional photonic crystal prepared from porous silicon (pSi) and containing the HCN-specific indicator dye monocyanocobinamide (MCbi), described previously[46]. In addition to its ability to selectively sense HCN vapors, the pSi sensor was chosen for this study because it is susceptible to interference from ambient environmental chemicals (Supplementary Fig. 1) that degrade its performance—providing a validated platform to test the function of our 2D protein crystal as a "gatekeeper". In the present system, the pSi substrate was modified with primary amines to impart a net positive surface charge and to

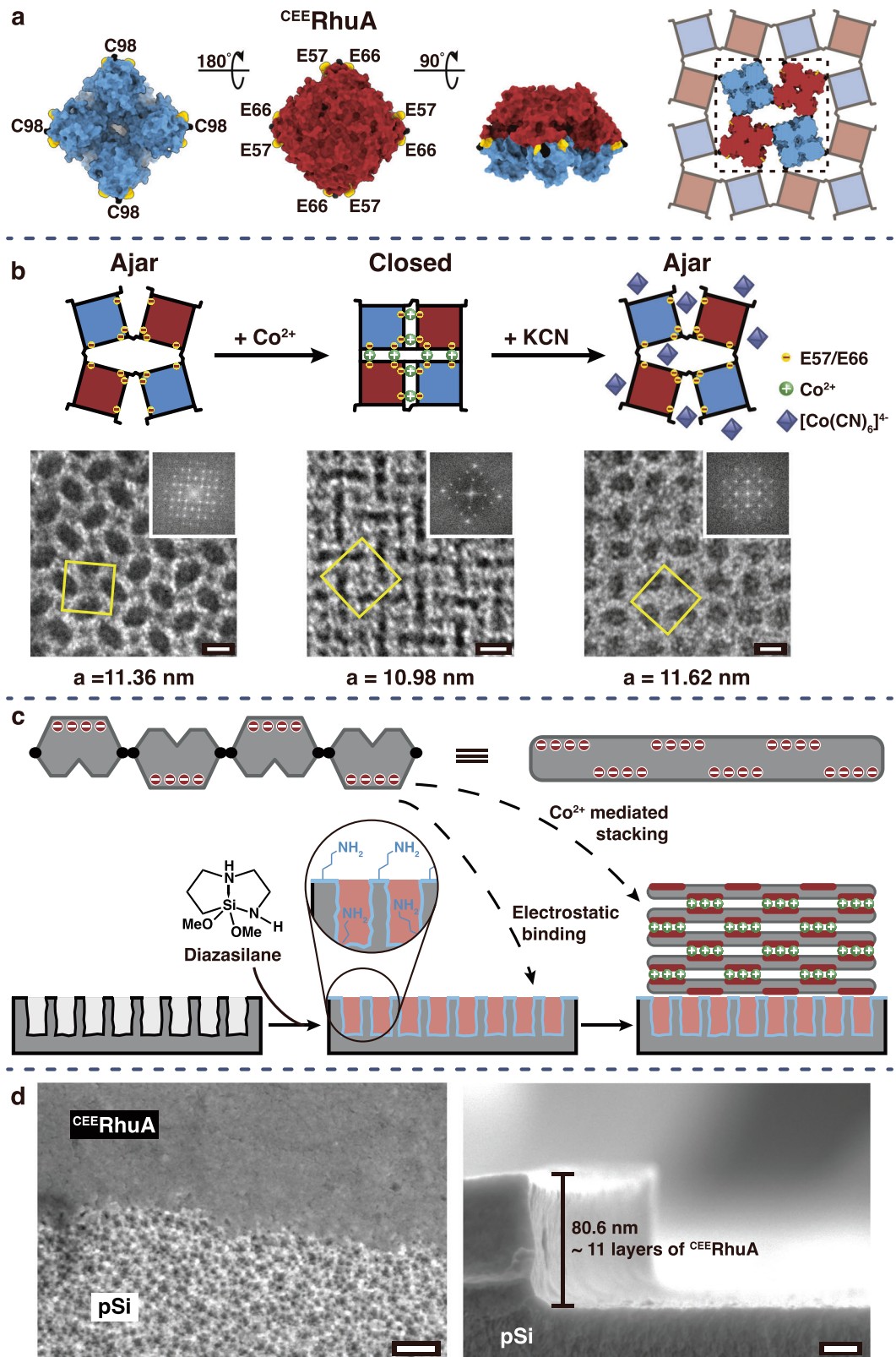

enable immobilization of the $^{CEE}$RhuA crystals onto the top of the pSi substrate via noncovalent electrostatic interactions. A cyclic azasilane reagent (2,2-dimethoxy-1,6-diaza-2-silacyclooctane; "diazasilane") was employed for this purpose, which reacts with Si-OH species at the surface of the pSi layer via a ring-opening reaction[47]. We also explored other amine-functionalized silanes, including the alkoxysilanes APDMES (3-aminopropyl(dimethyl)ethoxysilane) and APTES ((3-aminopropyl)triethoxysilane) and an heterocyclic azasilane (N-n-butyl-aza-2,2-dimethoxy-silacyclopentane), but found these to either yield weaker binding/coverage of the $^{CEE}$RhuA crystals or lead to the collapse of the surface pores and visible flaking (Supplementary Fig. 2). Ultimately, the diazasilane reagent 2,2-dimethoxy-1,6-diaza-2-silacyclooctane was selected due to its efficient reaction with hydroxy-terminated pSi and the lack of appreciable polymerization of the

**Fig. 1 | Design of the chemically triggered protein gatekeeper on a mesoporous silicon sensor. a** Renders of the protein gatekeeper [CEE]RhuA, highlighting its square shape, the locations of engineered cysteines at its corners (C98, in black) with proximal glutamate residues (E57/E66, in yellow) and the self-assembled 2D crystal topology, shown in the thermodynamically stable "ajar" state. The coloring indicates the net charge on each face of the [CEE]RhuA monomer; red indicates net negative surface charge density. **b** Simplified depiction of the dynamic [CEE]RhuA crystal motions that result from the addition of $Co^{2+}$ ion and then $CN^-$, alongside negative-stain transmission electron microscope (ns-TEM) images of the resulting [CEE]RhuA crystals (unit cell depicted as a yellow square overlaid on the image; 2D Fourier transform is shown in the upper right inset of each image; scale bars: 5 nm). The left image corresponds to the $Co^{2+}$-free "ajar" state, the central image to the $Co^{2+}$-bound "closed" state, and the right image to the "ajar" state that results from the interaction of the $Co^{2+}$-bound "closed" state of the crystal with $CN^-$. Addition of

$CN^-$ induces formation of the stable $Co(CN)_6^{4-}$ ion, displacing the $Co^{2+}$ bound to the E57/E66 residues to allow the 2D structure to open back to its "ajar" state. **c** Diagram showing deposition of the 2D protein crystals onto the mesoporous silicon (pSi) sensor chip. The net negative charge density on the [CEE]RhuA protein (red circles) enables electrostatic adhesion to the positively charged aminated pSi surface. The 2D crystals stack as multilayers on the chip; the process is assisted by excess $Co^{2+}$, which coordinate to the negative faces of the [CEE]RhuA crystals during deposition. **d** Plan-view (left) and cross-sectional (right) scanning electron microscope (SEM) images showing the boundary between a deposited multilayer of the protein and the pSi substrate. The top portion of the left image (labeled [CEE]RhuA) shows the protein overcoat, the lower portion shows an un-coated region of the pSi substrate, revealing the ~15 nm mesopores. The right image shows the edge of a deposited film (80.6 nm thick); corresponding to ~11 individual 2D crystal layers on the pSi surface. Scale bars: 100 nm (left image) and 20 nm (right image).

resulting siloxy species, which can lead to micro/mesopore clogging[47]. The presence of organic signatures assigned to the surface-grafted amine functionalities was confirmed by attenuated total reflectance Fourier transform infrared (ATR-FTIR) spectroscopy (Supplementary Fig. 2). The [CEE]RhuA crystals possess a $p42_12$ plane group symmetry in which the neighboring [CEE]RhuA proteins are arranged in an alternating up-down fashion with respect to the principal four-fold crystal symmetry axis (Fig. 1a). Each [CEE]RhuA protein monomer has one side (the C-terminal facet) that is rich in aspartic and glutamic acids, imparting a net negative charge density to that side. Because of the alternating up-down distribution of the monomers in the 2D crystal, both 2D crystal faces (which are equivalent) contain these negatively charged domains. We expected that these negatively charged domains would drive electrostatic adsorption of the layer of [CEE]RhuA crystals onto the pSi surface (Fig. 1c).

To validate this expectation, both amine-functionalized and untreated ("bare") pSi chips were drop-cast with $Co^{2+}$-bound [CEE]RhuA crystal suspensions (~5 μM protein concentration) in parallel and allowed to dry overnight, followed by two rinses with a buffer solution to eliminate weakly or non-adsorbed species. For consistency, both of the fabricated pSi chips contained loaded MCbi. Scanning microscopy (SEM) imaging revealed extensive coverage of the amine-functionalized pSi wafers with $Co^{2+}$-[CEE]RhuA crystals, whereas the untreated pSi substrates were bare, displaying only the characteristic ~15 nm pores of the pSi substrate (Supplementary Fig. 2). Partially coated samples, with a portion of the $Co^{2+}$-[CEE]RhuA layers removed, are shown in Fig. 1d. Cross-sectional SEM images showed 80-to-120-nm thick $Co^{2+}$-[CEE]RhuA layers coated on the pSi substrates (Fig. 1d). Given that a single 2D [CEE]RhuA crystal is approx. 7 nm thick, this observation indicates that the drop-casting procedure generated multilayers consisting of between 10 and 15 $Co^{2+}$-[CEE]RhuA crystalline sheets, likely promoted by the presence of excess metal ions in the buffer solutions that mediated interlayer stacking (Fig. 1c).

### Response of the gatekeeper-modified porous Si sensor to HCN

With the [CEE]RhuA layers successfully tiled onto the pSi sensor surface, we next sought to verify that the pSi sensor element could still detect HCN. Synthesized by controlled electrochemical etch into a polished silicon wafer[46,48–51], the pSi sensor element consisted of a complex stack of mesoporous silicon layers whose porosity was configured such that the resulting pSi photonic crystal displayed two specific resonances, known as stop bands (Fig. 2). These stop bands appeared in the white light reflection spectrum as peaks centered at 575 and 710 nm, and they acted as "Reference" and "Signal" channels, respectively, for the sensor[46,48–50]. The wavelength of the Signal channel was tuned−based on previously determined preparation conditions−to maximally overlap with the principal absorbance band (~580 nm) of CN-bound MCbi[46,52,53]. The absorbance maximum of the CN-free MCbi appears at 535 nm (Supplementary Fig. 3), so it did not significantly perturb the Signal or Reference channels of the photonic crystal. The

sensor "Response" was quantified as a ratio of the "Signal" channel intensity to the "Reference" channel intensity as defined in Eq. 2 (Methods). Our prior work had demonstrated that with this approach, the pSi-MCbi photonic crystal sensors detect HCN reliably at concentrations as low as 5 ppm within 10 min of vapor exposure[46]. After impregnation of the pSi photonic crystal sample with the MCbi indicator, the sensor element was coated with the [CEE]RhuA layers to complete the sensor-gatekeeper assembly (Fig. 2).

When exposed to 10 ppm of gaseous HCN in air of 50% relative humidity, CN-binding to MCbi manifested as a decrease in the intensity of the "Signal" channel ($I_{signal}$) relative to the "Reference" channel ($I_{Reference}$) due to an increase in light absorption by the dye, resulting in a decrease in the magnitude of the Response function (Fig. 2). These data indicated that HCN could trigger a response in the pSi sensor even when the sensor was coated with the protein gatekeeper, although it did not establish if the [CEE]RhuA protein layers were acting as a gatekeeper. To address this latter issue, we next looked at the ability of the [CEE]RhuA protein layers to gate the transport of small molecule interferents into the pSi sensor.

### Rejection of interferents by the protein gatekeeper: molecular dynamics simulations

A common issue for sensors designed to detect trace toxins is interference from naturally occurring chemicals in the environment, which are usually present at much higher concentrations than the target analyte. Interferents can generate false positives, false negatives, reduction in sensitivity, or zero-point drift (drift of the sensor's signal from its baseline, or resting state value, in the absence of the target analyte). The purpose of the protein gatekeeper in this work was to reject these molecular contaminants. Thus, with the HCN sensing capability of the gatekeeper-coated sensor assembly established, we next addressed the question of whether the opening and closing of the physical gaps in the 2D [CEE]RhuA crystals could impact molecular transport to act as gatekeepers.

To assess the theoretical possibility for such switchable permeability, we performed all-atom molecular dynamics (MD) simulations of infinitely periodic disulfide-linked RhuA lattices in the "closed" ($Co^{2+}$-bound), and "ajar" (post-HCN-exposure) states (see Fig. 1b) to emulate conditions on the sensor surface and modeled the transport of an ensemble of $n$-hexane molecules through the lattice. Details of the calculations are provided in the Methods section of the Supplemental Materials and in Supplementary Fig. 4. In brief, solution-equilibrated structures of the RhuA lattices, including their first hydration layer around the proteins (expected to exist in the presence of ambient humidity), were determined and held fixed, while hexane was introduced to one side of the lattice. The simulations indicated that a single layer of the 2D crystal showed a 4−8-fold higher hexane transport rate when in the ajar state (>1 hexane/ns) relative to the closed state (Supplementary Fig. 4). For computational economy, we only simulated a single layer of a RhuA crystal. Because the thickness of

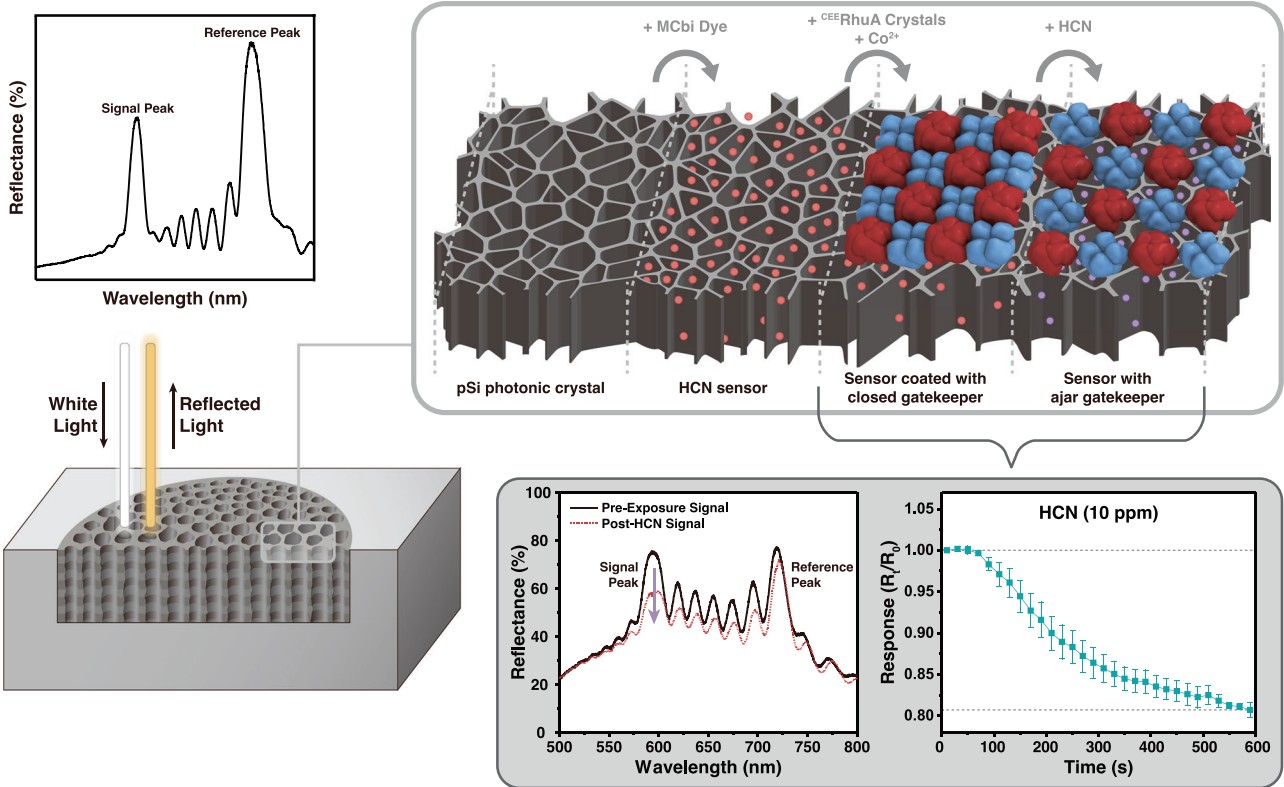

**Fig. 2 | Construction and operation principle of the gatekeeper-sensor assembly.** A porous silicon (pSi) photonic crystal is synthesized to display two resonances in the example optical reflectance spectrum, a signal peak, and a reference peak ("pSi photonic crystal"); an example plot for visualization is shown at left. A monocyanocobinamide (MCbi) indicator dye for HCN is then embedded in the porous layer. The wavelength of the Signal Peak coincides with the absorbance maximum of CN⁻ bound to MCbi (580 nm), while the Reference Peak appears in a wavelength region where no significant optical absorbance occurs, either before or after HCN exposure. The dye-loaded film is thus capable of remote optical detection of gaseous HCN ("HCN Sensor"). The $^{CEE}$RhuA 2D crystals are then layered over the top of the amine-modified pSi HCN sensor as described in Fig. 1C. At this point the gatekeeper is in its "closed" form. Introduction of gaseous HCN (10 ppm, in air of 50% RH) disrupts the $Co^{2+}$ links in the protein, causing it to open into its "ajar" state. HCN then enters the pSi layer, where it binds to MCbi. The resulting increase in optical absorbance of the MCbi-CN complex is detected as a decrease in the measured intensity of the "Signal Peak" relative to the "Reference Peak" in the optical reflectance spectrum. The sensor readout is quantified as the Response function defined in Eq. 2 of the methods section. Error bars are one standard deviation from the mean.

the gatekeeper coating on the pSi sensors consisted of 10 to 15 stacked layers of $^{CEE}$RhuA crystals, the aggregate multilayer coating can be expected to be substantially less permeable than what was calculated for the single layer in the closed state. Conversely, opening of the lattice pores in a single layer was expected to be sufficient to permit significant transport of gaseous species through a multilayered structure. This expectation was borne out in the experimental data, where no ingress of hexane through the closed layer was detected in the 10-min timescale of the measurements, whereas substantial infiltration was observed for the ajar state (vide infra).

### Sensing of HCN in complex matrices using the gatekeeper-modified pSi sensor

The gatekeeper-coated pSi sensor was challenged with a complex mixture of non-specific interferant vapors. A typical data trace is given in Fig. 3a, showing sequential addition of dry $N_2$, water vapor in air at 50% relative humidity (RH), a mixture of 43 different volatile organic compounds (VOCs) in air at 50% RH, and finally this same 43-VOC mixture, spiked with 10 ppm of HCN. The mixture of 43 VOCs used in this work was a composition of commonly occurring atmospheric chemicals that has been defined by the US Environmental Protection Agency (EPA) as "TO-14A" mixture[46]. The response function from the sensor showed little deviation from the zero point through the course of the experiments in the absence of HCN, and a substantial signal was recorded when the sensor was exposed to HCN. These results illustrate

that the above interferents did not impede the sensor's ability to respond to its target HCN analyte.

We next tested a range of more reactive chemicals, which might potentially degrade sensor operation due to specific reactions with the protein or indicator dye components. In separate experiments, sensors were exposed to hydrogen chloride (HCl, 100 ppm) ammonia ($NH_3$, 100 ppm), and n-hexane-saturated (3100 ppm) air (Fig. 3b). All the above potential interferants were delivered in humidified air streams of values 20, 50, and 80% relative humidity (RH). The sensors were exposed to the potential interferents for 10 min and the responses were compared with those obtained on separate, unexposed pSi chips using challenges of 5 and 10 ppm HCN vapor. As a reference, the HCN response to non-coated sensors was also evaluated. The (closed) gatekeeper-coated sensor showed no significant response to the vapor interferants at all values of RH tested, and it showed a similar response to HCN as compared to the non-gatekeeper-coated sensor (Fig. 3b). These results illustrate that the sensor did not give a false-positive signal from the potential interferents, and it performed reliably over the range of humidity values between 20 and 80% RH. Humidity conditions above and below these ranges were not evaluated in this study.

### Rejection of high concentrations of n-hexane by the gatekeeper

Being primary components of gasoline or diesel fuel fumes, aliphatic hydrocarbons are common interferents encountered in the field. For

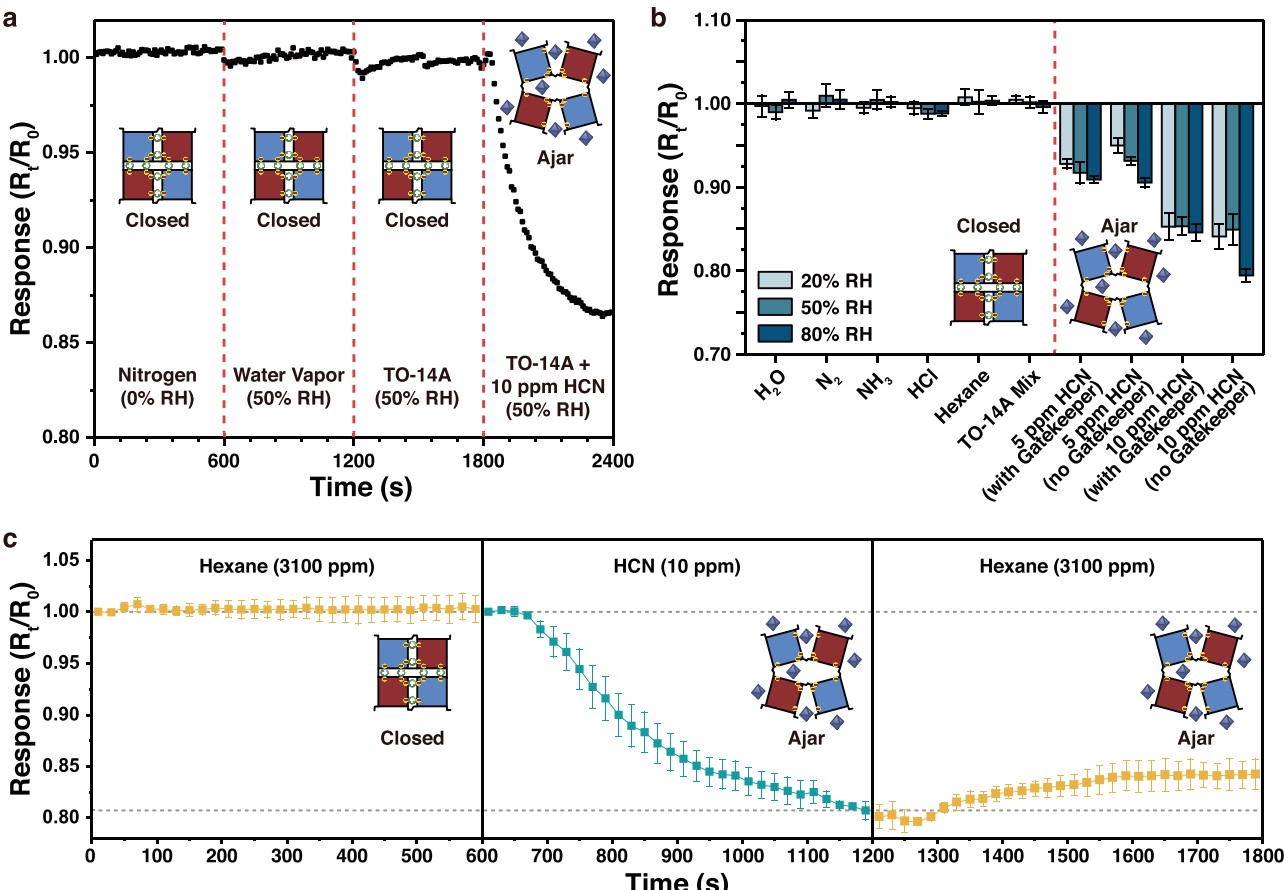

**Fig. 3 | Performance of ᶜᴱᴱRhuA-pSi sensors in the presence of common interferents. a** Response of the ᶜᴱᴱRhuA-pSi sensor to a series of potential interferents. The full sensor, consisting of the MCbi indicator dye embedded in the pSi photonic crystal, and coated with the $Co^{2+}$-ᶜᴱᴱRhuA crystal multilayer as shown in Fig. 2 is evaluated. Successive 10-min exposures of the sensor to a flowing stream of: nitrogen, air of 50% relative humidity (RH), an EPA TO-14A mixture of 43 common volatile organic compounds (1 ppm of each component) in air of 50% RH, and 10 ppm of HCN spiked into the same EPA TO-14A mixture in air of 50% RH. The sensor response is quantified as the $R_t/R_0$ response function as defined in Fig. 2 (relative to 0 s). Sensor shows negligible responses to all potential interferents but shows a strong response to the HCN analyte. **b**. Response values as in (**a**) but determined after 10 min of exposure to the indicated substance. All substances were diluted in flowing air to final concentrations of: $[H_2O]$ = RH at 22 °C as indicated; $[N_2]$ = 100%, water vapor added to obtain indicated RH at 22 °C; $[NH_3]$ = 100 ppm; $[HCl]$ = 100 ppm; [n-hexane] = 3100 ppm; TO-14A mixture in air. Error bars represent standard deviation from triplicate measurements on independently prepared samples. **c** Evaluation of sensor performance in the presence of a massive quantity of hexane vapor interferent. Trace shows successive 10-min exposures of a single sensor to a flowing stream of hexane-saturated air, followed by 10 ppm of HCN vapor in air, followed by hexane-saturated air. Relative humidity (RH) was maintained at 50% throughout the run. Response values as defined in (**a**). All traces are representative of triplicate measurements on independent samples. Error bars are one standard deviation from the mean.

the pSi sensor used in this study (in the absence of the ᶜᴱᴱRhuA gatekeeper layer), air saturated with n-hexane generated a substantial zero-point drift in the sensor (Supplementary Fig. 1). Thus, hexane-saturated air provided a good test of the gatekeeper's rejection capabilities. Nonpolar hydrocarbons also present a chemical environment that is significantly different from the aqueous environment where proteins normally function, so hexane-saturated air also provided a challenging medium to test the function of the protein gatekeeper.

As a test of the ability of the closed gatekeeper to reject n-hexane but to admit it upon entering the open "ajar" state, the sensor chip (MCbi-impregnated pSi photonic crystal with an 80 nm-thick $Co^{2+}$-ᶜᴱᴱRhuA gatekeeper top layer) was sequentially exposed to flowing n-hexane-saturated air, 10 ppm of HCN in air, then n-hexane-saturated air again (Fig. 3c). In the initial (closed) state, the sensor displayed no significant response to hexane-saturated air, consistent with the computational expectations and the experiments of Fig. 3b, which showed that the closed gatekeeper was not permeable to hexane. Opening of the gatekeeper upon exposure to HCN generated the expected response in the MCbi indicator; a large change in the $R_t/R_0$ response function was recorded during a 10-min exposure to the

analyte (Fig. 3c) that could not be reversed by flushing with a non-HCN air stream (Supplementary Fig. 5). Subsequent exposure of the sensor to n-hexane-saturated air generated a significant change in the measured signal that was opposite from the HCN response (Fig. 3c). Analogous sequential exposure experiments using $NH_3$, another hydrophilic metal-binding interferent, showed no shift in response under any conditions (Supplementary Fig. 6).

To understand the results of Fig. 3c, it must be stressed that the primary source of chemical specificity in the current sensor system is the strong and selective binding of $CN^-$ to the MCbi indicator dye, and the subsequent change in its absorbance spectrum. Hexane, as with most chemicals naturally existing in the atmosphere, does not bind to MCbi and thus does not generate such a colorimetric response, and it does not displace the strongly bound $CN^-$ from the indicator dye. Thus, the origin of the positive zero-point drift associated with hexane vapor in this system is not from its interaction with MCbi; rather, it derives from its non-specific adsorption within the pores of the pSi sensor. This produces a change in the average refractive index of the pSi film and in the index contrast in the photonic crystal, which results in a red shift of the stop bands and a decrease in their intensity

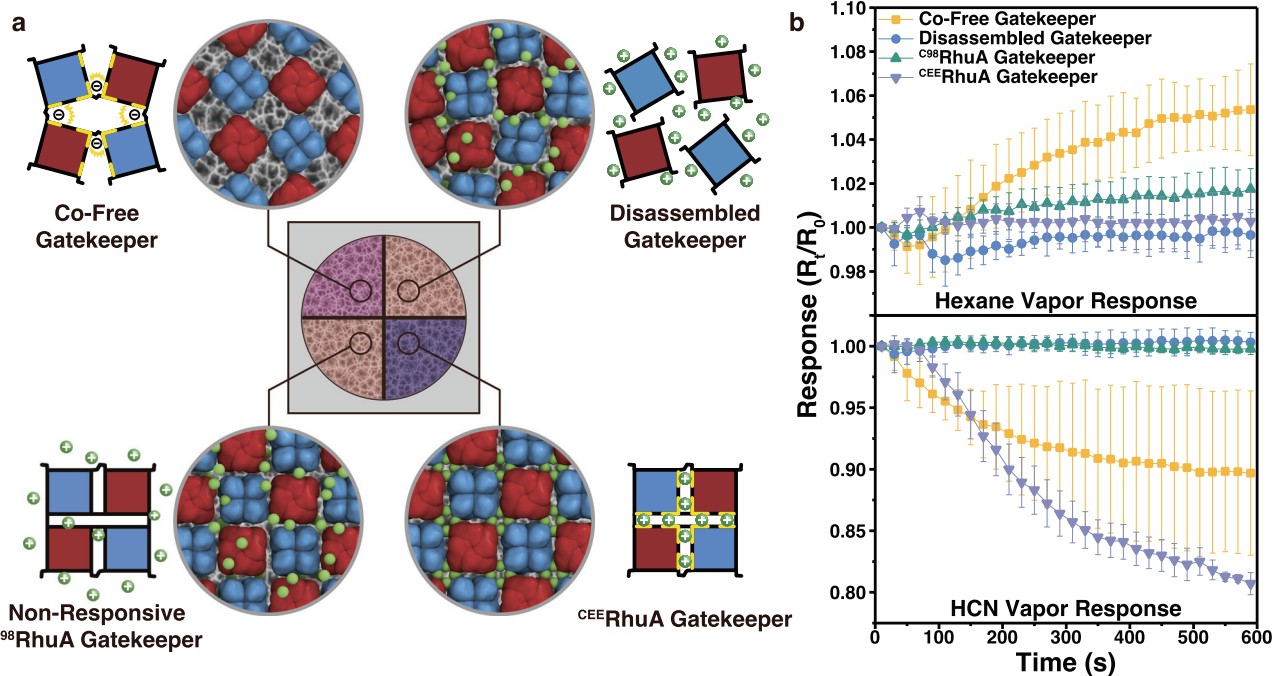

**Fig. 4 | Comparison of the CEERhuA gatekeeper with inactive protein control samples: response to n-hexane and to HCN. a** Diagram depicting the functional Co²⁺-CEERhuA-coated sensor in this study (upper left, "CEERhuA Gatekeeper") and control samples containing the same HCN-active pSi sensor element as in the functional sensor, but with different protein layers on the pSi sensor surface (clockwise, from upper right): "Disassembled Gatekeeper", an amorphous layer of CEERhuA protein monomers lacking disulfide bonds, co-deposited with Co²⁺ (expected to result in clogged pSi pores); "Co²⁺-free Gatekeeper", a layer of 2D CEERhuA crystals prepared with disulfide links but without Co²⁺ (an "always ajar" gatekeeper); "Non-Responsive C98RhuA Gatekeeper", a layer of 2D protein crystals using the C98RhuA variant of the protein (present in an "always closed" state). The small green spheres represent Co²⁺ ions; yellow bars represent glutamate residues E57/E66. All experiments were carried out on individually prepared chips. **b** (top) Response of the indicated samples to a flowing stream of n-hexane-saturated air (50% RH). $R_t/R_0$ response function is as defined in Eq. 2; the zero-point of the sensor is $R_t/R_0 = 1.00$. Sample response to n-hexane showed that the permanently ajar "Co²⁺-free Gatekeeper" is permeable, generating a positive zero-point drift in

response to hexane infiltration in the pSi nanostructure. The "CEERhuA Gatekeeper", the "Disassembled Gatekeeper" and the "Non-Responsive C98RhuA Gatekeeper" lattices were all impermeable and did not respond to n-hexane. Relative to "Co²⁺-free Gatekeeper", the closed forms exhibited statistically distinct responses (CEERhuA Gatekeeper, $p = 0.023$, Disassembled Gatekeeper, $p = 0.0085$, C98RhuA Gatekeeper, $p = 0.045$). (bottom) Response of the sensors to HCN. In contrast, exposure of the sensors to HCN triggered a strong response in the functional sensor ("CEERhuA Gatekeeper") and in the always-open "Co²⁺-free Gatekeeper" ($p = 0.067$); no significant response was detected in the "Disassembled Gatekeeper" ($p = 7.8 \times 10^{-6}$) and the "Non-Responsive C98RhuA Gatekeeper" ($p = 9.2 \times 10^{-6}$). The reported $p$ values are relative to the "CEERhuA Gatekeeper". Whereas the "Co²⁺-free Gatekeeper" permitted entry of both gaseous species, only the fully functional "CEERhuA Gatekeeper" sensor exhibited both HCN responsiveness and minimal zero-point drift (prior to HCN exposure). Data points represent averages of three independent measurements on independent samples. Error bars are one standard deviation from the mean. Statistical significance was obtained through Tukey's multiple comparison test (summarized in Supplementary Table 3).

(Supplementary Fig. 1)[46]. These effects are convoluted with the non-linearity of the spectral response of the spectrometer detector. While these phenomena exert relatively minor effects on the reflection spectrum of the pSi photonic crystal, at high concentrations of interferent they can overwhelm the sensor response to its target analyte[49]. In the present system, hexane-saturated air typically generated a +5% change in the zero-point of the sensor response function in the absence of the gatekeeper (Supplementary Fig. 1). As the absorbance peak of CN-bound Cbi is fixed (Supplementary Fig. 3), sufficiently large shifts in the centers of the reflectance peaks due to infiltration of fouling agents can result in artificially high $R_t/R_0$ values even in the presence of HCN. The consequence can be observed in Fig. 3c after HCN exposure, where subsequent hexane infiltration created a perceived change in the opposite direction of the signal from the HCN analyte; thus, it generated a false negative response in the pSi sensor when the gatekeeper was open, and no response when the gatekeeper was closed (Fig. 3c).

In addition to the information given by the response function $R_t/R_0$, which derived from changes in optical absorbance of the MCbi dye upon its binding of CN⁻, the stop bands also displayed a small red shift, whose magnitude scaled with concentration of hexane in the pSi layer. This wavelength shift is related to the average refractive index of the

pSi film, which is more dependent on the total mass of material within the pores than on the optical absorbance of the MCbi indicator dye included in the film, especially in the wavelength range of the "Reference" stop band, where the optical absorbance of MCbi is negligible[50]. Thus the red shift of the stop bands was used to confirm infiltration of the n-hexane vapor through the gatekeeper structures and into the mesoporous pSi chip (Supplementary Figs. 1, 6–9). The wavelength response to hexane vapors provided a useful in situ diagnostic of whether the gatekeeper was open or closed that was independent of the HCN sensing modality and confirmed that the gatekeeper was effective at blocking ingress of the organic vapors until it was triggered to open.

## Validation of the gatekeeper function

The basic gatekeeper function was validated with a series of control experiments, summarized in Fig. 4. The Figure compares the response of the complete CEERhuA gatekeeper-sensor assembly with three control samples, in which the composition of the protein "gatekeeper" coating was varied. The first control used a 2D protein crystal constructed from the C98RhuA variant of the protein ("non-responsive C98RhuA", Fig. 4a), which thermodynamically favors the closed state due to solvent entropy effects[40]. For the second control, the 2D

CEERhuA crystals were prepared in the absence of added Co²⁺ ("Co²⁺-free Gatekeeper", Fig. 4a), which generates the 2D lattice in its permanently "ajar" state as indicated in Fig. 1b. The third control sample was a "disassembled" form of the protein consisting of CEERhuA monomers that were lacking the disulfide bonds needed to bind their corners together, resulting in an amorphous protein layer when deposited onto the pSi chip ("disassembled gatekeeper", Fig. 4a). The addition of tris(2-carboxyethyl)phosphate (TCEP), a non-sulfurous reducing agent, to RhuA crystals results in complete dissolution into individual proteins (Supplementary Fig. 10). This latter structure thus was expected to contain no voids either before or after exposure to HCN and was considered as an example of a permanently closed gatekeeper.

The response of each of these controls, relative to the full gatekeeper-sensor assembly, to HCN and hexanes is shown in Fig. 4b. Response of the samples to a flowing stream of n-hexane-saturated air (50% RH) showed that the permanently ajar "Co²⁺-free Gatekeeper" control was permeable to n-hexane, generating a positive zero-point drift in response to hexane infiltration in the pSi nanostructure that differed significantly from the other samples, reaching a positive drift of >0.05 from the starting point. As noted above, the positive drift in the $R_t/R_0$ value following interferent infiltration can produce false negative signals in this sensor, highlighting the need for selective permeability. In contrast, the "CEERhuA Gatekeeper", the "Disassembled Gatekeeper" and the "Non-Responsive C98RhuA Gatekeeper" lattices were all impermeable and did not respond to n-hexane, with triplicate measurements within error of each other and remaining close to the initial zero-point of 1.00 (Fig. 4b). Consistent with the hexane response of each sample, exposure of the sensors to HCN triggered a strong response in the functional sensor ("CEERhuA Gatekeeper") and in the always-open "Co²⁺-free Gatekeeper", and no response in the "Disassembled Gatekeeper" and the "Non-Responsive C98RhuA Gatekeeper" (Fig. 4b). The HCN response was tightly distributed around the mean for all except the "Co-Free Gatekeeper", which exhibits large standard deviations in the Response curve, though it is distinct from the impermeable controls. It is possible that the hypothesized role of Co²⁺ in mediating interlayer stacking (Fig. 1c) is important for a coherent gatekeeper response on the surface, and its absence leads to less consistent deposition that could account for the wider variance of this specific control.

Of the above variants of the sensor, only the fully functional "CEERhuA Gatekeeper" sensor (as described in Fig. 1 and characterized in Fig. 3) exhibited both HCN responsiveness and minimal zero-point drift (prior to HCN exposure). Crucially, the lack of response for the "C98RhuA gatekeeper" demonstrates both that the presence of Co²⁺ alone is insufficient to produce a false-positive signal, and that conformational responsiveness of the crystal pores to metal binding is a requirement for selective permeability. Relatedly, the fact that the individual proteins can only occlude the surface but not selectively open in response to HCN (with all other aspects of the system remains the same; Fig. 4 and Supplementary Fig. 8) indicates that crystallinity is essential for the gatekeeper's function.

Additional control experiments were performed to verify that all components of the gatekeeper-integrated sensor were necessary for HCN-gated sensing via sequential exposure of hexane, HCN, and hexane. We also confirmed that NH₃, another metal-binding vapor, could not substitute directly for HCN. The findings are summarized in Supplementary Table 2 and Supplementary Figs. 1, 6–9. These data confirm that the CEERhuA gatekeeper in its closed form effectively rejects small molecule interferents such as hexane, and when triggered into its open form, it allows the ingress of all molecules, including the HCN analyte. We discovered a singular exception to this behavior for ethanol (an amphiphilic molecule found in exhaust fumes) at 80% RH where limited permeability could be inferred via peak shift (Supplementary Fig. 11), though this effect was not observed at lower %RH, and

no false-positive signal was observed at any conditions due to the selectivity of the MCbi dye. We characterized the effect of increasing ethanol concentrations on CEERhuA lattices in solution and confirmed that >10–20% (v/v) ethanol can induce lattice opening (Supplementary Fig. 12). These experiments indicated that despite the sensor robustness to a wide category of interferents, combined interactions between ethanol and water may disrupt the protein interfaces, which are also partially solvent-mediated[40]. Despite this narrow limitation, the key in achieving very high specificity in this system was the combination of specific interactions between HCN and the CEERhuA gatekeeper, and between HCN and the MCbi indicator dye.

In conclusion, this work demonstrates the successful incorporation of a genetically engineered and structurally well-defined protein-based assembly into a solid-state device and the exploitation of its unique structural and dynamic attributes. The use case in this study focused on a passive indicator for HCN gas that can be assessed remotely through optical interrogation while remaining impervious to ambient clutter. The rejection of non-HCN chemical species present in the environment was achieved through the flexibility of the CEERhuA crystals, which only became permeable upon loss of specific chemical interactions (metal coordination) that were directly severed by HCN molecules. As a result, the sensor demonstrated resistance to erroneous readouts due to a variety of non-specific interferents: changing relative humidity, HCl or NH₃ vapors in air, air saturated with hexane, and a complex mixture of VOCs (EPA TO-14A) in air, while displaying a response to low-ppm levels of HCN. The gatekeeping ability of the CEERhuA crystals was enabled by the specific interaction between the Co²⁺ metal ions and the carboxylate sidechains on the CEERhuA protein, which were engineered to maintain the crystals in their closed form upon binding to Co²⁺. The high affinity of Co²⁺ for HCN was key in displacing these protein-Co²⁺ bonds and triggering the protein into its open state, providing a closed/opened opening. The 2D structure of CEERhuA crystals was also important in that it enabled effective coverage of the pores in the sensor, blocking the ingress of interferent molecules that would normally cause an unacceptable zero-point drift in the sensor. Because the sensor transducer employed an optical readout, the transparency of the protein crystals in the visible region of the spectrum was also essential. Finally, the crystallinity of the protein mesostructure allowed it to exhibit its gatekeeper function in a non-liquid environment and immobilized on the surface of a Si chip, conditions that are generally not favorable for protein structure or function. Given that the structure and dynamics of the CEERhuA crystals were maintained through disulfide bonds with minimal footprints between the protein monomers, it should be possible to further engineer the protein surfaces such that the lattice opening/closing dynamics, and therefore the gatekeeping modality, can be rendered responsive to different stimuli. In addition, future investigations could further explore gatekeeper performance, stability, and long-term reliability through evaluating gatekeeper responses at lower limit concentrations of HCN, shelf-life studies as well as more challenging conditions such as high-humidity or low-humidity environments.

Proteins have featured prominently in functional devices and used in practical applications outside their native biological contexts, owing to their advantageous structural and physical properties, their functional specificities, and their ready accessibility to genetic and chemical manipulation[54–58]. Some notable examples of such applications include blood sugar monitors (glucose oxidase)[59,60], antigen detection kits (antibodies and derivatives)[61,62], and biosensors/DNA sequencing devices (membrane channels)[63–65]. In all these cases, this particular utility of a protein depends on its inherent structure and function as an individual unit. A major distinction of the present system is that it takes advantage of an emergent physical property (i.e., coherently dynamic 2D crystallinity) that is not present in the original protein building blocks but obtained through their self-assembly into a well-ordered material through properly chosen and placed chemical

linkages. This assembly-based approach establishes a paradigm for the incorporation of designed protein assemblies into solid-state devices as functional and dynamic components.

## Methods

### Materials

All materials and reagents were used as received. Single-crystal highly doped p-type (B-doped) silicon wafers of resistivity 0.8–1 mΩ·cm, polished on the (100) face, were purchased from Siltronix Corp. 2,2-dimethoxy-1,6-diaza-2-silacyclooctane [DMDASCP]) was purchased from Gelest Inc. Premixed hydrogen cyanide (HCN) gas cylinders (10 and 20 ppm; balance gas nitrogen) were purchased from Gasco Inc. EPA TO-14A Calibration mix was purchased from Restek through Linde Spectra Environmental Gases, catalog #226-34432 and contained 1 ppm of each component. All 43 components (1 ppm of each) are listed: acrylonitrile, 1,2-dichloroethane, toluene, benzene, 1,1-dichloroethene, 1,2,4-trichlorobenzene, bromomethane, cis-1,2-dichloroethene, 1,1,1-trichloroethane, 1,3-butadiene, 1,2-dichloropropane, 1,1,2-trichloroethane, carbon tetrachloride, cis-1,3-dichloropropene, trichloroethene, chlorobenzene, trans-1,3-dichloropropene, trichlorofluoromethane, chloroform, dichlorotetrafluoroethane, 1,1,2-trichlorotrifluoroethane, chloromethane, ethyl benzene, 1,2,4-trimethylbenzene, 3-chloropropene, 4-ethyltoluene, 1,3,5-trimethylbenzene, 1,2-dibromoethane, ethyl chloride, vinyl chloride, m-dichlorobenzene, hexachloro-1,3-butadiene, m-xylene, o-dichlorobenzene, methylene chloride, o-xylene, p-dichlorobenzene, styrene, p-xylene, dichlorodifluoromethane, 1,1,2,2-tetrachloroethane, 1,1-dichloroethane, tetrachloroethylene[66]. All other chemical reagents were purchased from Sigma-Aldrich.

### Instrumentation

Attenuated total reflectance Fourier transform infrared (ATR-FTIR) spectra were recorded on a Thermo Scientific Nicolet 6700 FTIR instrument fitted with a Smart iTR diamond ATRfixture. Scanning electron microscope (SEM) images were obtained with a Zeiss Sigma 500 in secondary electron imaging mode. Reflected light spectra were obtained in the visible-NIR spectral range with an Ocean Optics USB-4000 CCD spectrometer and a tungsten-halogen illumination source (Ocean Optics LS-1) connected with a Y-branch 600-µm-diameter, bifurcated multimode optical fiber. The common end of the bifurcated fiber was focused with an objective lens to a ~1 mm² spot and positioned with the optic axis normal to the sample surface. UV−vis absorbance spectra were obtained using a Molecular Devices SpectraMax 340PC384 Microplate Spectrophotometer with a 1 cm path length cuvette and 96-well microplate with normalized absorbance values to an equivalent 1 cm path length cuvette.

### Preparation of porous silicon (pSi) photonic crystals

Single-crystal highly doped p-type (B-doped) silicon wafers of resistivity 0.8–1 mΩ·cm, polished on the (100) face, were purchased from Siltronix Corp. and were cleaved into square chips of size 2 cm × 2 cm and mounted inside a custom Teflon etching cell with a 1.2 cm² exposed circular surface area to a 3:1 (v/v) 48% aqueous hydrofluoric acid:ethanol electrolyte solution (CAUTION: HF is highly toxic and can cause severe burns on contact with the skin or eyes). The wafer was contacted on the backside with aluminum foil and anodized with a platinum coil counter-electrode. Prior to the preparation of the porous layers with the photonic stop-bands, the samples were cleaned using a "sacrificial etch", which was used to etch a thin porous layer into the chip at 400 mA cm⁻¹ for 50 s. The porous layer was subsequently removed by dissolution with a strong base of 2 M aqueous KOH. The cell was then rinsed with water and ethanol three times, and fresh HF electrolyte solution was added to prepare two stop bands (dual peak pSi photonic crystal). The anodization waveform was generated using LabView software (National Instruments, Inc.), and the current was interfaced with a Keithley 2651 A Sourcemeter power supply. A composite time-dependent current density waveform, J(t), was created with the following Eq. 1 below:

$$J(t) = \left[\frac{J_{max} - J_{min}}{4}\right]\left[\sin\left(\frac{t}{T_1}\right) + \sin\left(\frac{t}{T_2}\right)\right] + \left[\frac{J_{max} + J_{min}}{2}\right] \quad (1)$$

With $J_{max} = 4.167$ mA/cm² and $J_{min} = 4.167$ mA/cm². For the preparation of the signal stop-band and reference stop-band, values of $T_1 = 8.4$ s and $T_2 = 9.8$ s were used. The composite waveform was applied for 300 s and the chips were subsequently rinsed with ethanol to remove any residual electrolyte.

### Surface functionalization of porous silicon (pSi) photonic crystals

Silicon chips containing the as-etched pSi photonic crystals were chemically modified by subjecting the chips to a cyclic organosilane reagent to functionalize the surface with amine (NH₂). The pSi photonic crystal chips were initially incubated in H₂O₂ (aqueous, 30% by mass) for 25 min at room temperature to generate a hydroxylated (Si-OH, Si-O-Si) porous Si layer as described previously[46]. The chips were washed by sequential triplicate rinses with 100% ethanol, 50:50 ethanol:dichloromethane, and 100% dichloromethane, and then submerged in a glass vial containing 1.5 mL of the heterocyclic silane (2,2-dimethoxy-1,6-diaza-2-silacyclooctane [DMDASCP]) for 3 h under gentle stirring. Excess silane was removed, and the chip was again subjected to serial triplicate rinses in 100% dichloromethane, then 50:50 ethanol:dichloromethane, then 100% ethanol, and the resulting amine-modified pSi-Si(NH₂) photonic crystal was allowed to dry.

### Synthesis and loading of monocyanocobinamide (MCbi)

Pure aquohydroxocobinamide (OH(H₂O)Cbi) was synthesized by base hydrolysis from hydroxocobalamin (Sigma-Aldrich) as described previously[46,53]. For this work, aquohydroxocobinamide was received as-is. A stock solution of monocyanocobinamide (CN(H₂O)Cbi, referred to here as "MCbi") was then prepared by reacting equimolar quantities of aquohydroxocobinamide (dissolved in aqueous 1 M NaOH) with KCN (Fisher Scientific) for 6 h under mild agitation at room temperature, which produced a dark red solution. A successful synthesis of MCbi was verified by identification of the characteristic absorbance peak spanning 475–520 nm using a benchtop UV-Vis spectrometer (Molecular Devices Spectramax 340Pc384 Microplate Spectrophotometer). The MCbi stock solution was diluted in ethanol until the concentration of the solution was 500 µM. A 100 µL aliquot of MCbi solution was subsequently drop-cast onto the amine-modified pSi-Si(NH₂) photonic crystal substrates and allowed to dry at room temperature for ~5 h or until the ethanol solvent appeared fully evaporated. Importantly, the amination chemistry did not clog the pores of the pSi layer; FTIR data confirmed that the amine-functionalized pSi photonic crystals were efficiently loaded with the MCbi indicator dye (Supplementary Fig. 2c).

### Synthesis, preparation, and deposition of gatekeeper crystals onto porous silicon (pSi) photonic crystals

CEERhuA protein was purified by published methods[40]. Briefly, CEERhuA was overexpressed in E. coli cells which were grown to high density (OD 0.6–1.0) and lysed by sonication. The resulting solution was clarified by centrifugation, precipitated by lowering the pH to 5, and, following the adjustment of the pH to 7.5, subjected to purification over a DEAE resin followed by an S resin, ultimately yielding approximately 10 mg of protein per liter of cell culture. The purified protein was concentrated to 150 µM (~18 mg/mL) in the presence of 10 mM β-mercaptoethanol and 1 mM ZnCl₂. About 0.5 mL of aliquots were flash-frozen in liquid nitrogen and stored at −80 °C. Crystallization was initiated by thawing the aliquots and leaving them on a shaking platform in a cold room at 4 °C for several days (up to several weeks).

CEERhuA crystal suspensions were clarified by repeated cycles of gentle centrifugation (~500 × g) in a benchtop centrifuge for 30–60 s, followed by substitution of the supernatant with fresh buffer solution containing 20 mM 2-(N-morpholino)ethanesulfonic acid (MES) at pH 6. 80–90% of the solution was replaced at each wash step, which led to complete exchange of the CEERhuA crystals into the new buffer solution. After washing, the crystal suspensions contained protein concentrations of ~100 μM. These suspensions were diluted to 10 μM and mixed 1:1 (v:v) with 20 mM MES buffer (pH 6) and 50 mM CoCl$_2$ and left at 4 °C for 3 days prior to deposition onto pSi chips. Experimental screening determined that a protein concentration of 5 μM provided functionally sufficient surface coverage of the sensor while preserving its optical reflectance spectrum. Crystal solutions were gently resuspended prior to deposition, using 50 μL to drop-cast onto 1.2 cm$^2$ square area chips and allowed to dry in air. To prepare partially coated samples shown in Fig. 1d, a 1.0 cm$^2$ rubber o-ring was used to isolate a circular deposition area for the CEERhuA crystal suspensions from the larger pSi substrate, and a scaled volume of protein suspension (~40 μL) was used for deposition. Once dried, the coated samples were cleaved to appropriate sizes to easily image the protein-sensor interface. Amine functionalization of the pSi surface (described above) was found to be an essential prerequisite for effective crystal binding to the substrate. The pKa of the amines in these chemical groups are estimated to be ~7.2, while the isoelectric point of the CEERhuA protein is ~5.5, so drop-casting of the protein crystals was carried out in the MES (pH 6) buffer introduced during the previous wash step, serving to promote binding by ensuring that the surface was positively charged while the protein remained negatively charged.

## Vapor dosing experiments

The photonic crystals were challenged with various streams of analyte and/or interferent vapors of fixed concentrations using mixed streams of dry, humidified compressed air, laboratory air, or nitrogen gas. All stream flow rates were controlled using digital mass flow controllers (MFC) that mixed the relevant compound(s) in their carrier gas with a relevant diluent gas stream, whose flow rate was set to achieve the desired component concentrations. The resulting vapors were introduced to the sample chamber through polyethylene tubing. Both HCN and EPA TO-14A vapors were produced from certified gas cylinders containing a known amount of the relevant vapor(s) in the air, and the concentration of the desired vapor was obtained by dilution with humidified laboratory air by means of the mass flow controllers. For water vapor and ammonia, aqueous solutions were prepared at a specified concentration and the vapors were generated by bubbling laboratory air through the liquids. Hexane vapor was generated by bubbling laboratory air through liquid hexane by means of a fritted glass gas dispersion tube to achieve saturation of the organic vapor in the laboratory air carrier gas. The concentrations of water, ammonia, and hexane in the carrier gasses were calculated using the relevant partial pressure values at 25 °C for the aqueous solutions of H$_2$O and NH$_3$ or for neat hexane, taken from literature values. The partial pressures were then converted to concentration values using previously described conversion methods[46]. In that calculation, P$_{analyte}$ is the partial pressure of the analyte vapor (Torr), F$_{analyte}$ is the flow rate of the analyte vapor stream (SCCM), and F$_{dilution,air}$ is the flow rate of the dilution air stream (SCCM). All RH values were verified using a probe hygrometer while sensing experiments were conducted with a net flow rate of 300 SCCM, controlled using an MFC. Reflectance spectra measured from the pSi sensors were collected using an Ocean Optics USB-4000 CCD spectrometer, coupled to one end of a bifurcated fiber optic cable. The two other branches, combined towards a single fiber, connected towards a tungsten filament light source and a collimating lens, fitted to the distal end and mounted 20 cm in front of the samples within its test chamber.

The Response function used to report the sensor data is defined as Eq. 2 below:

$$Response = \frac{R_t}{R_0}; R_t = \left(\frac{I_{signal}}{I_{reference}}\right)_t \& R_0 = \left(\frac{I_{signal}}{I_{reference}}\right)_{t=0} \quad (2)$$

where $R_t$ is the ratio of the intensity of the signal peak ($I_{signal}$) to the intensity of the reference peak ($I_{reference}$) measured at time $t$, $R_0$ is the ratio of the intensity of the signal peak ($I_{signal}$) to the intensity of the reference peak ($I_{reference}$) measured immediately before exposure (time $t=0$). The values of $I_{signal}$ and $I_{reference}$ were measured as the total number of counts measured in the reflection spectrum at the wavelength of the relevant peak. Therefore, a decrease in the value of the response function corresponded to an increase in absorbance from the CN-bound form of the MCbi indicator dye, associated with a positive HCN exposure. The wavelength (and relative shift in wavelength) of a given peak in the spectrum was calculated by measuring the wavelength at which the peak maximum appeared in the spectrum (in nm). No additional corrections for the instrument response function of the spectrometer were performed.

## Gas-phase simulations of hexane permeation through a single C98RhuA crystal layer

**Summary.** The all-atom molecular dynamics (MD) simulations employed infinitely periodic disulfide-linked RhuA lattices in the "closed" (Co$^{2+}$-bound), partially open ("ajar", post-HCN-exposure), and fully open states in vacuo to emulate conditions on the sensor surface (Supplementary Fig. 4). The two states of the lattice relevant to the present work are the "closed" and the "ajar" states; the fully open state is only observed with the pure protein crystals, in the absence of added Co$^{2+}$, but it was included in the simulations for comparison. Solution-equilibrated structures of these lattices, plus their first hydration layer (expected to exist in the presence of ambient humidity), were extracted and held fixed in the center of the simulation box, and two sequential additions (or "boluses") of neat hexane at 3130 ppm were added on one side of the lattice followed by equilibration. In the first bolus, a clear ordering of permeability from open to closed was observed, as monitored by a fraction of hexane molecules below the lattice after 25 ns of equilibration, with very few molecules permeating through the closed gatekeeper. The calculations indicate that many of the hexane molecules adsorbed to the gatekeeper surface, as reflected in "lag times" of ~1, 2.5, and 4.5 ns for open, ajar, and closed states, respectively, so a second equivalent bolus was added to the system. After this second round of equilibration, the ajar and open states were fully permeable to hexanes, while the closed state was observed to be >2-fold less permeable (as a single layer). The transport rates for the open and ajar states (>1 hexane/ns) are 4–8-fold higher than that for the closed state.

**Computational details.** Coordinates for pre-equilibrated infinitely periodic C98RhuA crystal lattices at the three different opening angles (open, ajar, closed) were extracted from simulations described previously[67]. Briefly, those simulations were run using NAMD 2.13[68] using the CHARMM36[69] forcefield and TIP3P water in standard three-dimensional periodic boundary conditions (3D PBC) at a temperature of 300 K at constant pressure (NPT), where the xy dimensions were held fixed, and the volume was allowed to fluctuate along the z dimension. This enabled the crystal lattice conformation to be maintained in a realistic fashion. These coordinates were lightly re-equilibrated to ensure uniform preparation for the simulations described herein. The configurations for open, ajar, and closed were re-solvated in water boxes of dimensions: a = b = 200.000 Å, 178.885 Å, 151.186 Å, respectively; c = 114.0 Å for all simulations (leaving ~40 Å of total solvent separating repeating layers along the z periodic dimension), with neutralizing sodium ions. Protein coordinates were held

fixed during a 2000-step gradient descent minimization of the solvent and ions. This was followed by equilibration and production sampling using the same simulation parameters described above. Nonbonded interactions were truncated at 12.0 Å, with a smooth switching initiated at 10.0 Å, a margin parameter of 5.0, and 2.0 fs timestep, and Langevin thermostat and barostat with Particle Mesh Ewald (PME) for long-range electrostatics. The protocol consisted of a 20 ps equilibration of the water to stabilize the pressure and volume along the $z$ periodic dimension, followed by a 2 ns equilibration and 3 ns of production sampling. During equilibration and production, harmonic restraints were applied to help maintain planarity and symmetry in the simulation box as follows: the entire center of masses (COM) for two of the diagonally opposed monomers were restrained to $z = 0$ (define an absolute $z$ position for the rest of the lattice), the COMs of each pair of diagonally opposed monomers in each individual RhuA tetramer were constrained to have a difference in $z$ coordinate of 0 (to keep each tetramer balanced in the plane while allowing free diffusion along $z$), and the $xy$ COM-COM distances of each pair of diagonally opposed tetramers were constrained to have a difference of 0 (to ensure crystal symmetry). Force constants were all 100 kcal/mol/Å². From the final frame of the production sampling, the $^{C98}$RhuA proteins and any waters whose oxygen was within 4.0 Å of the surface (the first hydration layer)[40] were extracted to represent the protein lattice coordinates for permeation simulations.

Atom types for hexane molecule coordinates were assigned using the standard alkane parameters in CGenFF (part of CHARMM36). It was estimated that the experimental apparatus exposed the sensors to approximately 3100 ppm hexane, close to other estimates of the density of gaseous hexanes at 25 °C of approximately 3200 ppm. We computed that, given the $xy$ dimensions of each lattice and an assumed fixed z dimension of 120 Å, a total of 105, 84, and 60 hexane molecules would achieve a uniform 3130 ppm of hexane in the gas phase. PACKMOL was used to generate uniform distributions of these molecules to these $xy$ box dimensions and a slightly shorter 110 Å $z$ dimension, with the knowledge that upon merging with the protein crystals, some additional void volume would be introduced that would bring the hexane density to 3000–3100 ppm. These hexane configurations were equilibrated for 10 ns at constant volume and 300 K. The last frame was taken for use.

Finally, the equilibrated hexanes were positioned above the equilibrated $^{C98}$RhuA coordinates. The lattice was moved along $z$ such that the average alpha carbon position was at $z = 0$. All protein atoms were held fixed for the duration of the simulations. Now with a constant $z$ dimension of 300.0 Å (Supplementary Fig. 4), we employed NAMD 2.14 with the Multiscale Summation Method[70] to enable 2D PBC conditions, where periodicity was enforced along $xy$ and while being nonperiodic along $z$. Harmonic constraints using the TclBC module of NAMD were employed to keep all hexanes within the $z$ dimensions. An MSM grid spacing of 5.0 and MSMQuality of 1 were used for all simulations. Equilibration was carried out for 25 ns, at which point the final frame was extracted and merged again with the equilibrated hexanes to generate the second "bolus" of hexane and approximate steady-state hexane exposure on one side of the lattice. The positions of the hexane molecules were computed using an ad hoc Tcl script, which counted the number of hexane carbon atoms above and below specific $z$ values, then divided by 6 to reduce quantization error. Hexane permeation simulations were all carried out in triplicate from the same starting coordinates but randomized initial velocities.

## TEM and SEM imaging
For the preparation of negative-stain TEM samples, 3.5 μL aliquots of protein crystals were applied onto negatively glow-discharged Formvar/carbon-coated Cu grids (Ted Pella, Inc.), washed with 50 μL of filtered milli-Q water, and stained with 3.5 μL of filtered 2% uranyl acetate solution. Grids were imaged using JEOL 1400 plus transmission

electron microscope operating at 80 keV, equipped with a Tungsten filament and a bottom-mounted Gatan OneView (4k × 4k) camera or FEI Sphera transmission microscope operating at 200 keV, equipped with a LaB6 filament and Gatan 4 K CCD. Unit cell parameters of the protein 2D lattices were estimated from the Fast Fourier Transformation of the captured real space images in the program Fiji (http://fiji.sc/Fiji).

For the preparation samples for SEM imaging, pSi chips were prepared as described above. For plan-view imaging, prepared chips were cleaved to a size of 1 cm × 1 cm and applied onto a 12.5 mm diameter SEM pin stub mount (Ted Pella, Inc.) and held in place with conductive carbon tape (Ted Pella, Inc.). For cross-sectional imaging, the prepared chips were cleaved to a size of 1 cm × 0.5 cm, with a portion of the chip not containing any protein crystals. These samples were then mounted on a low-profile 90° SEM stub mount (Ted Pella Inc.) with the cleaved portion of the chip flush with the top of the SEM mount. All samples were then sputtered-coated with Ir using a K575X Sputter Coater (Emitech) for a single 5-min cycle. The samples were then imaged using a Sigma 500 (Zeiss Group, USA) Field Emission SEM operating at 15 kV.

## Illustration of protein structures
Molecular graphics images were rendered using UCSF ChimeraX[71], developed by the Resource for Biocomputing, Visualization, and Informatics at the University of California, San Francisco.

## Data availability
The principal data supporting the findings of this work are available within the figures and the Supplementary Information. Additional data that support the findings of this study are provided in the Source Data Files. Source data are provided with this paper.

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

## Acknowledgements

The research is based upon work primarily supported by NSF through the UC San Diego Materials Research Science and Engineering Center (UCSD MRSEC), grant DMR-2011924. Additional support received from the Office of the Director of National Intelligence (ODNI), Intelligence Advanced Research Projects Activity (IARPA), via contract number 2018-18071700005. The views and conclusions contained herein are those of the authors and should not be interpreted as necessarily representing the official policies or endorsements, either expressed or implied, of the ODNI, IARPA, or the US Government. The US Government is authorized to reproduce and distribute reprints for Governmental purposes, notwithstanding any copyright annotation thereon. Specifically, the contract provided support for S.V., R.G.A., and Y-S.L. Additional funding was provided by DOE (Division of Materials Sciences, Biomolecular Materials, Award DE-SC0003844, for the initial work on dynamic 2D RhuA crystals; CSSAS–The Center for the Science of Synthesis Across Scales–under Award Number DE-SC0019288, for TEM imaging). The authors acknowledge the use of facilities and instrumentation supported by the National Science Foundation through the UC San Diego Materials Research Science and Engineering Center (UCSD MRSEC) DMR-2011924, and by the San Diego Nanotechnology Infrastructure (SDNI) of UCSD, a member of the National Nanotechnology Coordinated Infrastructure, which is supported by the National Science Foundation (Grant ECCS-2025752). S.V. acknowledges financial support from the Natural Sciences and Engineering Research Council of Canada Postgraduate Scholarship—Doctoral program (NSERC PGS-D). Molecular graphics in Figs. 1a, 2, 4, and S4 are performed with UCSF ChimeraX, developed by the Resource for Biocomputing, Visualization, and Informatics at the University of California, San Francisco, with support from the National Institutes of Health R01-GM129325 and the Office of Cyber Infrastructure and Computational Biology, National Institute of Allergy and Infectious Diseases.

## Author contributions

S.V. and Y-S.L. developed and assembled all tested porous silicon constructs and ran the vapor dosing experiments. R.G.A. and Z.Z. prepared the protein constructs and the solution-grown crystals. S.V., Y-S.L., Z.Z., and R.G.A. performed all analytical experiments and imaging. S.V., R.G.A., and Z.Z. performed all data processing and analysis. R.G.A. performed the simulations and theoretical analysis. A.C. and S.V. synthesized the MCbi dye; F.A.T., M.J.S., and G.R.B. provided experimental guidance. F.A.T., M.J.S., C.E.W., J.S.H., and D.E.H. conceived and oversaw the project. R.G.A., S.V., Z.Z., M.J.S., and F.A.T. prepared the figures and wrote the manuscript.

## Competing interests

MJS is a scientific founder (SF), member of the Board of Directors (BOD), Advisory Board (AB), Scientific Advisory Board (SAB), acts as a paid consultant (PC) or has an equity interest (EI) in the following: Aivocode, Inc (AB, EI); Bejing ITEC Technologies (SAB, PC); Hinalea Imaging (EI); Lisata Therapeutics (EI); Illumina (EI), Matrix Technologies (EI); NanoVision Bio (SAB, EI); Precis Therapeutics (SF, BOD, EI), Quanterix (EI), Spinnaker Biosciences, Inc. (SF, BOD, EI); TruTag Technologies (SAB, EI); and Well-Healthcare Technologies (SAB, PC). Although one or more of the grants that supported this research has been identified for conflict-of-interest management based on the overall scope of the project and its potential benefit to the companies listed, the research findings included in this publication may not necessarily relate to their interests. The terms of these arrangements have been reviewed and approved by the University of California, San Diego, in accordance with its conflict-of-interest policies. The remaining authors declare no competing interests.
