## [Peer Review File · Nature Communications]

Designed 2D Protein Crystals as Dynamic Molecular Gatekeepers for a Solid-State DeviceREVIEWER COMMENTS

Reviewer #1 (Remarks to the Author):

The paper by Vijayakumar et al. proposes the integration of a dynamic 2D crystalline protein assembly, CEERhuA, with a mesoporous silicon (pSi) photonic crystal-based remote sensor as a dynamic gatekeeper for the selective admission and detection of hydrogen cyanide (HCN). 2D CEERhuA crystals are engineered via metal coordination bonds to undergo a substantial and coherent conformational change from a closed state (pore dimensions <1 nm) to an ajar state (pore dimensions ~4 nm) in the presence of HCN. Although the paper is very interesting, it is extremely difficult to believe that a protein has a 2D structure like graphene-flake. I would suggest the authors to clarify at which extension this protein complex is 2D. Moreover, the paper seems more fundamental than applicative/integration as stated in the abstract. Indeed, the authors are exploring the mechanism of such interactions with CN⁻ ions. In terms of sensing devices there are plenty of CN⁻ ions sensors. It is not clear from the manuscript the revolutionary approach of the proposed platform.

Comments:

1. In the abstract, the acronym CEERhuA should be written at least the first time in its extended version.
2. Please reformulate this sentence 'For the present work, we induced the closed state of CEERhuA through the agency of the divalent metal ion Co²⁺ (Fig. 1b)'. Simply it could be modified 'For the present work, we induced the closed state of CEERhuA by using Co²⁺ as divalent metal ion.'
3. The authors should present some results proving the different kinetics in the formation of complex both with Ca²⁺ and Co²⁺. Why the kinetics is so slow for Co²⁺ complex formation (3 days)? Did the author investigate the chemical-physical parameters that may this equilibrium?
4. Which is the linear range? Is it an ON-OFF sensor? Did the authors test the device in real samples? All analytical figures of merit are missing.
5. A table of comparison with other papers for the detection of HCN should be reported.

Reviewer #2 (Remarks to the Author):

The manuscript by Vijayakumar and coworkers describes a method to establish selectivity of HCN sensing by a Si photonic crystal sensor. The authors have agglomerated their previous work on silicon sensors and a protein crystal engineering approach to form a "gatekeeper" which eliminates responses of the detector to other interfering analytes. It is an interesting approach to the problem of selectivity and is presented in a logical manner. While the manuscript illustrates well the approach and the authors have correctly considered some of the possible interfering agents such as water (humidity) and VOCs, there are several issues that the authors should address prior to publication:

- 1) There are commercially available HCN sensors which operate at the sensitivity of the presented gatekeeper sensor but the authors have not attempted to compare or contrast the relative characteristics of their sensor with commercially available ones.
- 2) Why has hexane (and VOCs) been used as the possible interferant? These may be common environmental pollutants affecting the operation of sensors but I think HCN sensors ought to be checked for more relevant interferents such as carbon monoxide or hydrogen. Given the construction of the gatekeeper system, a lack of response to those analytes would be a more significant feature.
- 3) Related to (2), hexane and components of TO-14A are uniformly hydrophobic. What happens to the gatekeeper layer under hydrophilic vapors? Humidity is mentioned as an interfering factor but other common interferents could be methanol and other alcohols especially in the automotive (gasoline/diesel) context mentioned by the authors.
- 4) The interferents HCl and ammonia are always delivered under humid conditions where they might be sequestered in the vapor phase compared to HCN. What happens under RH = 0 % conditions? Also, do the data shown at the right side of Figure 3b refer to HCN sensing by a sensor after it has been exposed to HCl, NH₃, etc.?
- 5) While HCN is referred to as a chemical warfare agent, I found this label misleading in the title since there are other better known agents such as sarin, mustard gas, etc. The detection of HCN is

much more likely to be required in an industrial context since it is commonly used as a reagent.
6) There is no discussion about the stability and/or durability of the sensor in use.

Overall, I found the work an effective demonstration of the use of the authors' concept, although the authors have not included the effects of more difficult possible interfering analytes.

Reviewer #3 (Remarks to the Author):

The manuscript titled: Designed 2D Protein Crystals as Dynamic Molecular Gatekeepers for Selective Sensing of a Chemical Warfare Agent, presents a combination of two previously developed concepts (pSi sensor and responsive protein 2D crystals) to form a protected sensor for Co reactive compounds. The figures are informative, and the manuscript is mostly well-written. The concept is interesting and an important effort towards practical application of the RhuA 2D crystals. However, the most important claims of the work are currently not properly justified by the provided data. Detailed protein structural characterization is provided only in solution and correlation between degree of protein crystallinity vs. sensing performance has not been established. The biggest challenge with the manuscript is that the opening and closing of the protein lattice has not been convincingly demonstrated on the pSi surface. Also, it is not clear how selective the sensor is in the presence of other Co binding agents.

1. Since Co also binds the 2D protein crystals on the pSi, are they peeled off by the addition of CN (in solution).
2. Page 4, line 129: How fast is the opening? How is the solution opening kinetics compared to the kinetics on surface?
3. 150 mM KCN is rather high considering the analytical motivation of the work. What is the lowest possible concentration of KCN required for opening? Later in the paper only 10 ppm CN is used.
4. Page 4, line 130-132: it is mentioned that also simple removal of the Co is enough to open the lattice. In this case, what is the rationale of adding 150 mM KCN if pure water will also induce the opening?
5. This also brings up the question of selectivity. How can the authors be sure that KCN is responsible for opening if dialysis or mixing also causes the opening? How about other ions that can efficiently scavenge or displace Co?
6. How does the opening/closing work when the 2D protein layers are bound to the surface? It would be surprising if the opening is as efficient inside a multilayer structure. The conformational change on surface must be proven when only 10 ppm HCN is added.
7. The TO-14A composition and used concentration is not sufficiently described in the manuscript or ref. 46.
8. Figure 3a and b should show control samples where the uncoated sensor has been first polluted with eg. N₂ and then TO-14A + HCN is added, and no response is observed. This would indicate that the gatekeeper actually helps in preventing sensor fouling.
9. Related to Figure 4: How were the different protein films characterized and controlled to be similar / comparative. I would assume that e.g. coating thickness plays a major role.
10. Figure 4b data lacks quantitative discussion. Are the observed differences statistically significant?

REVIEWER COMMENTS

REVIEWER #1

General Comments: The paper by Vijayakumar et al. proposes the integration of a dynamic 2D crystalline protein assembly, ^{CEE}RhuA, with a mesoporous silicon (pSi) photonic crystal-based remote sensor as a dynamic gatekeeper for the selective admission and detection of hydrogen cyanide (HCN). 2D ^{CEE}RhuA crystals are engineered via metal coordination bonds to undergo a substantial and coherent conformational change from a closed state (pore dimensions <1 nm) to an ajar state (pore dimensions ~4 nm) in the presence of HCN. Although the paper is very interesting, it is extremely difficult to believe that a protein has a 2D structure like graphene-flake. I would suggest the authors to clarify at which extension this protein complex is 2D. Moreover, the paper seems more fundamental than applicative/integration as stated in the abstract. Indeed, the authors are exploring the mechanism of such interactions with CN⁻ ions. In terms of sensing devices there are plenty of CN⁻ ions sensors. It is not clear from the manuscript the revolutionary approach of the proposed platform.

Response: We thank Reviewer 1 for their comments and suggestions. In response, we have attempted to clarify the significance of this work in our revision. In addition to our responses to the specific comments below, we would also like to address these general comments.

First, the protein building block itself is not two-dimensional per se; rather, its self-assembled structure is a 2D crystal, as depicted in **Figure 1a** and imaged by TEM in **Figure 1b** and by SEM in **Figure 1d**. This work directly builds from several previous publications involving these self-assembled RhuA lattices, including Suzuki, Y. et al. *Nature* (2016) [10.1038/nature17633] (Ref. 36), Alberstein, R.G. et al. *Nature Chemistry* (2018) [10.1038/s41557-018-0053-4] (Ref. 40), and Zhang, S. et al. *Nature Communications* (2020) [10.1038/s41467-020-17562-1] (Ref. 70), with the 2018 publication introducing the 2D ^{CEE}RhuA lattices (which is the subject of the current work). We refer the reviewer to these manuscripts for extensive characterization into the properties of these 2D crystals. If the reviewer is referring to the RhuA crystals not being as “thin” as a single-layer graphene, we would like to point out that dimensionality of an object is associated with the aspect ratios of its width, depth, and height (i.e., its x, y, and z dimensions). Any object whose x, y dimensions are much larger than its z dimension are technically called two-dimensional (for example, a sheet of paper is a two-dimensional object). If, on the other hand, one goes by absolute dimensions, then even a very thin object like a single-layer graphene would technically not be two-dimensional, because every object has a volume (i.e., a non-zero height, width and depth).

Second, we do not report any fundamental investigations into the nature of Co²⁺-CN⁻ interactions in this manuscript precisely because this work is applicative in nature and not fundamental. We instead cite existing knowledge regarding the stability of cyano-cobalt compounds in Ref. 45 (Gail, E. et al. in *Cyano Compounds*, Inorganic. In Ullmann’s Encyclopedia of Industrial Chemistry) as our reasoning for the use of Co²⁺ ions together with previously described metal-dependent ^{CEE}RhuA pore-size changes from Alberstein et al. (2018). All described measurements in this work that are not imaging-based are instead spectroscopic measurements of visible light interactions with the porous silicon substrate, giving direct readouts for sensing performance under real-world conditions.

Finally, as correctly stated by the Reviewer, there are already many existing CN⁻ sensors and we do not claim superior performance as a particular goal or property of the described sensor. The differentiating aspect of the present work is less about HCN sensing specifically. Rather, it is about the demonstration that a designed protein assembly can be used to gate analyte transport in a highly selective manner. This not only constitutes a novel application for self-assembled protein materials, but also demonstrates that this functionality can occur outside of the aqueous environment in which proteins typically exist, which is a largely unexplored topic. These advancements are stated at the beginning and end of the Conclusions section: “*This work demonstrates the successful incorporation of a genetically engineered and structurally well-defined protein-based assembly into a solid-state device*

and the exploitation of its unique structural and dynamic attributes.” and “A major distinction of the present system is that it takes advantage of an emergent physical property (i.e., coherently dynamic 2D crystallinity) that is not present in the original protein building blocks but obtained through their self-assembly into a well-ordered, extended material through properly chosen and placed chemical linkages. This assembly-based approach establishes a paradigm for the incorporation of designed protein assemblies into solid-state devices as functional and dynamic components.”

Comment 1: In the abstract, the acronym ^{CEE}RhuA should be written at least the first time in its extended version.

Response: We have adopted the Reviewer’s suggestion for improved clarity and changed the sentence introducing CEERhuA in the Abstract to read: “Here we report the integration of a dynamic 2D crystalline protein assembly of C98/E57/E66 L-rhamnulose-1-phosphate aldolase (^{CEE}RhuA), with a mesoporous silicon (pSi) photonic crystal-based remote sensor as a dynamic gatekeeper for the selective admission and detection of hydrogen cyanide (HCN).” We further define it again for clarity in the Introduction: “This study addresses these questions by integrating a dynamic, 2D protein crystalline material self-assembled from the C98/E57/E66 variant of L-rhamnulose-1-phosphate aldolase (^{CEE}RhuA) recently engineered in our laboratories as an analyte-triggerable membrane into a porous silicon (pSi)-based, remote sensing chip.”

Comment 2: Please reformulate this sentence ‘For the present work, we induced the closed state of ^{CEE}RhuA through the agency of the divalent metal ion Co²⁺ (Fig. 1b)’. Simply it could be modified ‘For the present work, we induced the closed state of ^{CEE}RhuA by using Co²⁺ as divalent metal ion.’

Response: We thank the Reviewer for this suggestion. We have adopted the proposed change and the sentence in the Introduction now reads: “For the present work, we induced the closed state of ^{CEE}RhuA using Co²⁺ as a divalent metal ion (Fig. 1b).”

Comment 3: The authors should present some results proving the different kinetics in the formation of complex both with Ca²⁺ and Co²⁺. Why the kinetics is so slow for Co²⁺ complex formation (3 days)? Did the author investigate the chemical-physical parameters that may this equilibrium?

Response: The kinetics of Ca²⁺ binding is not relevant for this work, as we opted to directly test with Co²⁺ due to its ability to interact/form a stable complex with CN⁻ (as noted in our response to the General Comments above; Ca²⁺ does not form stable complexes with CN⁻). Thus, any difference in kinetics is not applicable to the functionality of the reported sensor. Nevertheless, it was reported in Alberstein et al. (2018) that Ca²⁺ induces full closure of ^{CEE}RhuA crystals over an approximately three-day period, so the binding kinetics of Co²⁺ is likely to be comparable. In the aqueous environment in which binding occurs, metal hydration and complexation with ^{CEE}RhuA surface glutamates (both hard oxygen-based ligands) are likely to be comparable chemical environments, potentially explaining the slow binding kinetics of both cations. Regardless, three days is sufficient to achieve equilibrium closure of the ^{CEE}RhuA crystals in the presence of Co²⁺ ions, and thus was used as a consistent length of time to ensure that deposited crystals were all in the closed conformation. As this work did not focus on the fundamentals of ^{CEE}RhuA-Co²⁺ interactions, these kinetics were not investigated further.

Comment 4: Which is the linear range? Is it an ON-OFF sensor?

Response: Dose-response curves were not used in this work. The analyte dosing concentrations were determined from References 46 (Lu, Y.-S. et al. *ACS Sensors* (2021)) and 48-50 for the nanostructured

porous silicon photonic crystal and Reference 43 (Greenawald, L.A. et al. *Sens Actuators B Chem* (2015)) for the indicator dye. This prior work ensured that the sensor would yield a strong spectroscopic response to enable confident assessment of the gatekeeper function.

Comment 5: Did the authors test the device in real samples?

Response: Yes. For example, the data in **Figures 3a and 3b** were performed in laboratory air or laboratory air spiked with interferents commonly encountered in real world environments as discussed in the text in subsection “*Sensing of HCN in Complex Matrices using the Gatekeeper-Modified pSi Sensor*”.

Comment 6: All analytical figures of merit are missing.

Response: This is an important point, although it is somewhat outside the scope of the present study. The objective of this paper was to use a sensing system with well-established readout to report on whether the gatekeeper was open or closed under different conditions. In the context of assessing performance of the gatekeeper, the relevant analytical figures of merit are in the form of a yes-no question—is the gatekeeper open to allow molecular transport, or is it closed, thus blocking transport on the timescale of the experiments? The Reviewer brings up an important point for performance of the gatekeeper, namely, at what lower limit of concentration of cyanide does the gatekeeper begin to open? We did not address this question in the present paper, but it is an important one for future study. We have added a statement to this effect to the conclusions section: “*In addition, future investigations could further explore gatekeeper performance, stability, and long-term reliability through evaluating gatekeeper responses at lower limit concentrations of HCN, more complex environmental condition, and shelf-life studies.*”

Comment 7: A table of comparison with other papers for the detection of HCN should be reported.

Response: We thank the Reviewer for this suggestion as it provides important context for our work. We have added a table to the SI (**Extended Data Table S1**) describing relevant references, sensing modalities, and thresholds for HCN detection. We have also added the following sentence to the Introduction to clarify that our reported advance focuses on protein-inorganic integration into a sensor and not on sensor performance specifically: “*While many sensors exist for cyanide detection (Extended Data Table S1), we surmised that this pSi modality would provide an ideal platform for determining whether a protein gatekeeper could function in a controllable manner outside of its native aqueous environment, as the flat substrate is geometrically compatible with 2D crystal deposition and its established sensitivity to cyanide infiltration provided a clear readout of the porosity of the protein layers.*”

REVIEWER #2

General Comments: *The manuscript by Vijayakumar and coworkers describes a method to establish selectivity of HCN sensing by a Si photonic crystal sensor. The authors have agglomerated their previous work on silicon sensors and a protein crystal engineering approach to form a "gatekeeper" which eliminates responses of the detector to other interfering analytes. It is an interesting approach to the problem of selectivity and is presented in a logical manner. While the manuscript illustrates well the approach and the authors have correctly considered some of the possible interfering agents such as water (humidity) and VOCs, there are several issues that the authors should address prior to publication:*

Response: We thank Reviewer 2 for their positive comments and concrete suggestions for improving the manuscript.

Comment 1: There are commercially available HCN sensors which operate at the sensitivity of the presented gatekeeper sensor but the authors have not attempted to compare or contrast the relative characteristics of their sensor with commercially available ones.

Response: Although the distinguishing characteristic of our design is the selectivity bestowed by the gatekeeper layer (as opposed to claiming superior performance in sensitivity), this is an important point to address. We have added a table to the SI (**Extended Data Table S1**) of HCN sensors to present comparative characteristics of other works to this one.

Comment 2: Why has hexane (and VOCs) been used as the possible interferant? These may be common environmental pollutants affecting the operation of sensors but I think HCN sensors ought to be checked for more relevant interferents such as carbon monoxide or hydrogen. Given the construction of the gatekeeper system, a lack of response to those analytes would be a more significant feature.

Response: We chose to use hexane and other VOCs due to their prevalence in industrial environments where one might typically find HCN gas. We acknowledge the reviewer's point that CO or H₂ could be other important interferents to test, as they are also commonly found in industrial environments. However, we note that CO or H₂ are not known to form stable complexes with Co²⁺ (both require lower oxidation states of Co) and they would not be expected to open the gatekeeper. Furthermore, we feel that our battery of tests against interferents (including the complex VOC mixture TO-14A, composition reproduced below and now in the Methods as well), and hydrophilic vapors such as HCl and NH₃ (both capable of binding or outcompeting Co²⁺) as well as N₂ covers a wide variety of potential chemical properties that encompass what CO or H₂ would add. In any case, following the Reviewer's other comment below, we have now tested the effect of ethanol as an amphiphilic interferent is also found in environments such as combustion products.

Comment 3: Related to (2), hexane and components of TO-14A are uniformly hydrophobic. What happens to the gatekeeper layer under hydrophilic vapors? Humidity is mentioned as an interfering factor but other common interferents could be methanol and other alcohols especially in the automotive (gasoline/diesel) context mentioned by the authors.

Response: The Reviewer raises an important point: we agree that broad-spectrum rejection is an important demonstration for the reported system. As mentioned above, other hydrophilic interferents (including HCl and NH₃) were already tested and reported within the manuscript (**Figure 3b**). Following the reviewer's suggestion, we have now added ethanol as another interferent that is relevant in combustion contexts. We found that under conditions from very high ethanol concentrations (low relative humidity) to a 50:50 mixture of neat ethanol and water vapor, there continues to be no response to this interferent (**Extended Data Fig. S11**). Surprisingly, however, we found that at 80% RH there is a small peak shift implying induced porosity of the layer. We followed up these experiments with solution-phase characterization of the effect of increasing ethanol concentrations in aqueous buffer and determined by TEM that sufficiently high quantities (>10-20% v/v) of ethanol in solution could actually induce opening of the lattices (**Extended Data Fig. S12**). While investigating the mechanism of this effect is far outside the scope of this proof-of-concept manuscript, one hypothesis is that there may be some sort of combined interactions between ethanol and water that disrupts the interfaces (which are also partially solvent-mediated as described in Alberstein 2018, Ref. #40). We believe the fact that this is the only tested interferent condition to lead to a shift, while still not producing any response, does not

detract from the main conclusions of this work, but can be cited as an example of a limitation for future exploration.

We have addressed this point in the main text just before the conclusions as: *“We discovered a singular exception to this behavior for ethanol (an amphiphilic molecule found in exhaust fumes) at 80% RH where limited permeability could be inferred via peak shift (Extended Data Fig. S11), though this effect was not observed at lower %RH, and no false-positive signal was observed at any conditions due to the selectivity of the MCbi dye. We characterized the effect of increasing ethanol concentrations on ^{CEE}RhuA lattices in solution and confirmed that >10-20% (v/v) ethanol can induce lattice opening (Extended Data Fig. S12). These experiments indicated that despite the sensor robustness to a wide category of interferents, combined interactions between ethanol and water may disrupt the protein interfaces, which are also partially solvent-mediated.⁴⁰ Despite this narrow limitation, the key in achieving very high specificity in this system was the combination of specific interactions between HCN and the ^{CEE}RhuA gatekeeper, and between HCN and the MCbi indicator dye.”*

The new figures are reproduced below for the reviewer’s convenience.

Extended Data Fig. S11. Spectroscopic measurements of pSi sensors coated with a ^{CEE}RhuA gatekeeper in response to ethanol. Top: Reflectance spectra of pSi sensor obtained before and after a 10-minute exposure to ethanol at humidity values of 20%, 50%, and 80% RH. At 20% and 50% RH conditions, the sensor shows no noticeable change, however at 80% RH there is a small peak shift implying induced porosity of the layer. Bottom: the lack of change in the normalized signal response ($I_{\text{signal}} / I_{\text{reference}}$) are shown upon exposure to the ethanol vapor

Extended Data Fig. S12. Negative-stain TEM images of solution-state ^{CEE}RhuA gatekeeper in response to ethanol. Negative-stain transmission electron microscope (ns-TEM) images of the resulting ^{CEE}RhuA crystals following exposure to varying concentrations of ethanol. All percentages are reported as v/v. Between 1-10% ethanol, there are no obvious changes in the pore opening, though high molar concentrations of >10% can cause gatekeeper opening and eventually damage to the proteins themselves. It is possible that on the pSi surface, ethanol alone does not negatively affect the gatekeeper, but sufficiently high concentrations of water forming a mixture with the ethanol vapor may result in structural changes to the protein gatekeeper that render it unable to fully reject interferents. The insets are magnified versions of the regions highlighted in yellow to facilitate visualization of the pore conformational state. The scale bars are 100 nm.

Comment 4: *The interferents HCl and ammonia are always delivered under humid conditions where they might be sequestered in the vapor phase compared to HCN. What happens under RH = 0 % conditions?*

Response: This experiment was not done as 0% RH is not a realistic condition. However, the experiments in this work addressed this with minimal humidity at 20% RH, which is already quite dry on a real-world scale. We have added a comment to this effect in the main text: “In this work, conditions of very high and very low humidity values were not studied, as these extremely dry or extremely humid conditions are outside of the range encountered in most real-world scenarios”.

Comment 5: *Also, do the data shown at the right side of Figure 3b refer to HCN sensing by a sensor after it has been exposed to HCl, NH₃, etc.?*

Response: No. As stated in the Figure 3 caption: *“b. Response values as in (a) but determined after 10 min of exposure to the indicated substance”*. Thus, the right side (past the red dotted line) shows that HCN is capable of reaching the sensing layer underneath, with comparable response to the uncoated sensors, even in the presence of gatekeeper, while the data to the left of the dotted lines show that interferents themselves do not cause responses themselves. In contrast, **Figure 3c** shows that after opening of the gatekeeper with HCN in the second step, additional hexane could then reach the sensor layer and cause drift of the response that is not possible prior to HCN exposure.

For further clarity, we now report sequential exposure of hexane, NH₃, and hexane, to demonstrate the lack of permeability for another hydrophilic metal-chelating interferent. This has been added as **Extended Data Figure S6** (reproduced on the right) and now briefly discussed in the main text as part of our initial discussion of the Figure 3c results: “Analogous sequential exposure experiments using NH₃, another hydrophilic metal-chelating interferent, showed no shift in response under any conditions (Extended Data Fig. S6).”

Extended Data Fig. S6. Spectroscopic measurements of pSi sensors coated with a CEE RhuA gatekeeper in response to hexane and NH₃. Top: White light reflectance spectra of Co-bound-gatekeeper-coated pSi sensors, showing total stability of the spectra before and after sequential exposure of hexane, ammonia, hexane. Bottom: the Response panels correspond to those seen in the analogous experiment (using HCN instead of NH₃) in **Figure 3c**, and the Peak Shift panels depict the actual movement of the reflectance spectra that occurs upon interferent infiltration (as seen in **Extended Data Fig. S1**). Notably, the use of NH₃ as a substitute for HCN does not produce any peak shift in the 2nd hexane exposure, demonstrating that the gatekeeper remains firmly closed, in contrast with the data in **Figure 3c** for HCN. Error bars represent the standard deviation from triplicate trials.

Comment 6: *While HCN is referred to as a chemical warfare agent, I found this label misleading in the title since there are other better-known agents such as sarin, mustard gas, etc. The detection of HCN is much more likely to be required in an industrial context since it is commonly used as a reagent.*

Response: We agree with the Reviewer and have accordingly changed the title to: “Designed 2D Protein Crystals as Dynamic Molecular Gatekeepers for a Solid-State Device”. This title eliminates the

mention of chemical warfare agents and draws more attention to the advancement reported in this work, namely the use of protein crystals as responsive gatekeepers in a solid-state device.

Comment 7: There is no discussion about the stability and/or durability of the sensor in use. Overall, I found the work an effective demonstration of the use of the authors' concept, although the authors have not included the effects of more difficult possible interfering analytes.

Response: We thank Reviewer 2 about the supportive comment, and we agree that long-term stability is an important feature for a sensing device. At the same time, we feel that our work provides sufficient proof-of-concept for demonstrating that protein-based materials can function in a hard-device setting. Future work should include long-term stability in the environment and storage (shelf life) stability as properties of interest. We have added a sentence to the conclusions to this effect: "*In addition, future investigations could further explore gatekeeper performance, stability, and long-term reliability through evaluating gatekeeper responses at lower limit concentrations of HCN, more complex environmental conditions, and shelf-life studies.*"

REVIEWER #3

General Comments: The manuscript titled: Designed 2D Protein Crystals as Dynamic Molecular Gatekeepers for Selective Sensing of a Chemical Warfare Agent, presents a combination of two previously developed concepts (pSi sensor and responsive protein 2D crystals) to form a protected sensor for Co reactive compounds. The figures are informative, and the manuscript is mostly well-written. The concept is interesting and an important effort towards practical application of the RhuA 2D crystals. However, the most important claims of the work are currently not properly justified by the provided data. Detailed protein structural characterization is provided only in solution and correlation between degree of protein crystallinity vs. sensing performance has not been established. The biggest challenge with the manuscript is that the opening and closing of the protein lattice has not been convincingly demonstrated on the pSi surface.

Response: We thank Reviewer 3 for the supportive comments and insightful criticisms. On-surface structural characterization of the protein lattices is highly challenging and despite our attempts (for example, using GISAXS), we were not able to obtain high-quality data. However, we note that we have already included a control experiment (the "disassembled gatekeeper" in **Figure 4**) that clearly shows the importance of crystallinity for sensing performance: namely, when the crystals are reduced with TCEP (which breaks the disulfide linkages and dissociates the crystals), the resulting free proteins end up forming a permanently impermeable barrier to the pSi surface, likely due to pore clogging. In all of our experiments included in the study, the HCN-specific gatekeeper opening/sensing is only observed with intact ^{CEERhuA} protein crystals, providing convincing evidence that their (1) crystallinity, (2) opening/closing dynamics, and (3) their selective HCN-response are necessary for chemically triggerable gatekeeping.

To clarify the interpretation of this important control, we have added a new SI figure showing the above-described effect of TCEP (**Extended Data Figure S10**; reproduced below). We have also added the following to the main text during discussion of the controls: "*The addition of tris(2-carboxyethyl) phosphate (TCEP), a non-sulfurous reducing agent, to RhuA crystals results in complete dissolution into individual proteins (Extended Data Fig. S10).*" We later add in the discussion of **Figure 4b**: "*Of the above variants of the sensor, only the fully functional ^{CEERhuA} Gatekeeper" sensor (as described in Figure 1 and characterized in Figure 3) exhibited both HCN responsiveness and minimal zero-point drift (prior to HCN exposure). Crucially, the lack of response for the ^{C98RhuA} gatekeeper" demonstrates*

both that the presence of Co^{2+} alone is insufficient to produce a false-positive signal and that conformational responsiveness of the crystal pores to metal binding is a requirement for selective permeability. Relatedly, the fact that the individual proteins can only occlude the surface but not selectively open in response to HCN (with all other aspects of the system remaining the same; **Fig. 4** and **Extended Data Fig. S8**) indicates that crystallinity is essential for the gatekeeper's function."

Extended Data Fig. S10. Negative-stain TEM images of $^{CEE}RhuA$ gatekeeper in response to TCEP. The addition of 10 mM TCEP, a non-sulfurous reducing agent, to assembled $^{CEE}RhuA$ gatekeeper crystals (left) results in their complete dissolution into RhuA proteins (visible as individual square-shaped units on the right) by breaking the disulfide bonds that hold together the 2D lattices. The scale bars are (from left to right): 500, 200, 500, 200 nm, respectively.

General Comments (continued): Also, it is not clear how selective the sensor is in the presence of other Co binding agents.

Response: Other analytes such as NH_3 and HCl that could disrupt Co-protein interactions—and now ethanol—have all been tested and shown to result in no effect on the sensor performance (with the exception of neat ethanol in 80% RH, which may operate by solvation effects rather than metal interactions). Please see the above discussion in Reviewer 2's comments (particularly 2, 3, and 5), the new Extended Data Fig. S6 for sequential hexane-ammonia-hexane response, and the new changes in the text referring to the results of ethanol cited above in Reviewer 2 Comment 3.

Comment 1: Since Co also binds the 2D protein crystals on the pSi, are they peeled off by the addition of CN (in solution).

Response: This is an interesting question. TEM images of Co-bound crystals often show stacking that can be more excessive than $^{CEE}RhuA$ crystals without Co^{2+} ions. Addition of KCN in solution can lead to some apparent reduction of thickness, but this effect is inconsistent, and is not clearly observed in the TEM images shown below. It is possible that interlayer Co^{2+} ions are too sequestered by the protein layers to be effectively bound/stripped by CN^- anions. In the low-dielectric environment of air for on-chip testing, the strengthened electrostatic interactions between the acidic residues between the RhuA

Negative-stain TEM images of Co^{2+} -bound $^{CEE}RhuA$ gatekeeper in response to KCN. Zoomed-out images (scale bars: 1 μm) of gatekeeper crystals in response to the addition of 6 molar equivalents of KCN (relative to Co^{2+} ions). There is no clear trend in terms of reduction (or exacerbation) of stacking of crystals prior to deposition. It is possible that these ions are sufficiently sequestered that addition of KCN does not lead to delamination, and the lower dielectric environment of air during on-chip testing will likely strengthen the protein-metal interactions.

layers and the Co^{2+} ions may be even stronger, and the absence of a solution environment means that delamination is unlikely on the pSi surface.

Comment 2: Page 4, line 129: How fast is the opening? How is the solution opening kinetics compared to the kinetics on surface?

Response: Gatekeeper opening kinetics in solution are not available in a continuous fashion (it is difficult to measure this experimentally) and not directly relevant for the sensor in the intended ambient air environment. The measurements shown in **Figures 2 and 3** clearly indicate that the solid-state sensor responds within 3 min (when the change in intensity has reached $\sim 50\%$ of the full range). However, to address the reviewer's question, we did conduct time-course TEM imaging in solution (see figure on the right) after addition of KCN and observed that lattice opening could occur in as quickly as 2 mins, consistent with the approximate lag-time in response observed from our spectroscopic measurements on the pSi surface in **Figure 3c**.

Negative-stain TEM images of Co^{2+} -bound $\text{C}^{\text{EE}}\text{RhuA}$ gatekeeper in response to KCN. Zoomed-in images (scale bars: 100 nm) of gatekeeper crystals in response to the addition of 6 molar equivalents of KCN (relative to Co^{2+} ions) at $t=0$ min (a) and $t=2$ min (b) depict clear changes in gatekeeper porosity after KCN treatment after 2 minutes, which is comparable to the lag-time response to gaseous HCN on the pSi sensor surface (**Figure 3c**).

Comment 3: 150 mM KCN is rather high considering the analytical motivation of the work. What is the lowest possible concentration of KCN required for opening? Later in the paper only 10 ppm CN is used.

Response: We thank the reviewer for highlighting this ambiguity. The high concentration in aqueous solution is needed to titrate down the free Co remaining in the solution—this is just an experimental necessity. We have now clarified in the section “*Design of the 2D Protein Crystal Gatekeeper*” that 150 mM KCN was used to ensure that all Co^{2+} in solution was reacted to a 6:1 stoichiometric ratio to ensure full sequestration of the ions: “*A mixture of Co^{2+} -bound $\text{C}^{\text{EE}}\text{RhuA}$ crystal suspensions 1:1 (v:v) with a buffered 150 mM KCN solution (final $\text{CN}^-:\text{Co}^{2+}:\text{C}^{\text{EE}}\text{RhuA}$ molar ratio was 6:1:0.0002 to fully sequester free all Co^{2+} ions, both solvated and gatekeeper-bound) led to the coherent opening of the crystals and restoration of the original unit cell dimension (Fig. 1b).*” Once the $\text{Co}^{\text{CEE}}\text{RhuA}$ crystals are deposited on the pSi surface, there is not excess solvated Co^{2+} present, allowing significantly lower concentrations of KCN to cause the opening of the lattices. While determination of a lower limit is left to future work (as described above and now elaborated on in the conclusions), the clear effect of only 10 ppm HCN on the pSi sensors established that their response is appropriate for analytical motivations presented in the manuscript.

Comment 4: Page 4, line 130-132: it is mentioned that also simple removal of the Co is enough to open the lattice. In this case, what is the rationale of adding 150 mM KCN if pure water will also induce the opening?

Response: Co^{2+} removal from the protein crystals is not simple: the Co^{2+} ions bound to the protein crystals are not only stably bound (in a thermodynamic sense) but they are also kinetically stabilized. Their removal requires shear forces, extensive dialysis over long periods of time, or strong Co-complexing agents such as EDTA and, of course, CN^- . Placement of Co^{2+} -bound protein crystals in water is not sufficient to drive Co^{2+} removal in reasonable timescales due to the kinetic stabilization of Co^{2+} . An appropriate analogy can be made to any metalloprotein (i.e., a protein with a bound metal cofactor): while the placement of any metalloprotein in pure water should favor the demetallation of the

protein due to thermodynamics, the demetallation does not occur due to the kinetic stability of the metal ion bound within the protein scaffold.

Comment 5: This also brings up the question of selectivity. How can the authors be sure that KCN is responsible for opening if dialysis or mixing also causes the opening? How about other ions that can efficiently scavenge or displace Co?

Response: Reviewer 3 has a good point. Following from the preceding comment, dialysis and mixing represent chemical/physical processes that are not present on the sensor surface. That those strong driving forces can cause opening themselves simply demonstrates that the opening of gatekeeper pores is determined by whether or not they are bound to metal ions. In the applicative context (in air, on a solid surface), selectivity derives from the ability of the ligand (CN⁻) to form stable coordination complexes with the metal ion (Co²⁺) chosen to link together the proteins in the crystal lattice. As the Reviewer 3 correctly states and as we mention in our response to Reviewer 2 (Comment 2), there may be other analytes besides CN⁻ that that could form stable Co complexes or disrupt Co-protein bonds and lead to gatekeeper opening. Our control samples include NH₃ and HCl as potential disruptors of Co-protein bonds, but these analytes do not display any gatekeeper opening or colorimetric response. We can foresee that an analyte like H₂S could potentially chelate away the Co²⁺ ions and also give a colorimetric read-out with the cobinamide sensor unit. However, we provide sufficient and compelling proof-of-principle to demonstrate the mode of operation of our system and believe that a broader testing of analyte scope/specificity is better left for future studies.

Comment 6: How does the opening/closing work when the 2D protein layers are bound to the surface? It would be surprising if the opening is as efficient inside a multilayer structure. The conformational change on surface must be proven when only 10 ppm HCN is added.

Response: As we noted in an earlier response, this is a good point that we tried to address with GISAXS and TEM experiments, but the sensitivity and resolution of the techniques in our hands were not sufficient to provide a conclusive answer. However, our extensive control experiments (**Figure 4**) show that only crystalline, Co²⁺-bound, ^{CEE}RhuA crystals exhibit the anticipated behavior. Without Co (“Co-Free Gatekeeper”), the crystals are sufficiently porous to allow HCN infiltration. In the case of ^{C98}RhuA crystals+Co²⁺ (“Non-Responsive Gatekeeper”), their lack of switchability renders them impermeable under all conditions (they are permanently closed). In the case of RhuA proteins alone (“Disassembled Gatekeeper”, see also new **Extended Data Fig. S10**), the pSi pores become clogged, generating an analyte-impermeable barrier. The latter two controls also demonstrate that Co alone is not creating some false signal. The conclusion of these specific tests is that the gatekeeper must be crystalline (controllable porosity) and have a switch mechanism to effect pore size changes. While we were also pleasantly surprised that the conformational changes of the crystals indeed occur on a solid surface, this is likely because the surface adsorption method is noncovalent in nature, and the low dielectric environment strengthens the repulsive interactions between the glutamate residues (that bind Co²⁺ ions) upon Co²⁺ loss. The hydration layer(s) on the proteins (stemming from ambient humidity) may also contribute to lattice mobility. We note that over the course of HCN exposure, the response is gradual with time, rather than a discrete on/off response, which is also consistent with a gradual opening process that occurs over continuous exposure. We further refer the Reviewer to their comments 8 and 10 below where we describe greater elaboration on the interpretation of the results of **Figures 3 and 4** to help communicate these conclusions.

Comment 7: The TO-14A composition and used concentration is not sufficiently described in the manuscript or ref. 46.

Response: The full composition of TO-14A used in this work is given in the following citation and shown in the table on the right. We have added this information to the Materials subsection in the Methods.

Manning, J. A.; Burckle, J. O.; Hedges, S.; McElro, F. F. Compendium Method TO-14A. Compendium of Methods for the Determination of Toxic Organic Compounds in Ambient Air 2nd ed.; Environmental Protection Agency: 1999.

TO-14A Components of Interest (1 ppm of each)		
acrylonitrile	1,2-dichloroethane	toluene
benzene	1,1-dichloroethene	1,2,4-trichlorobenzene
bromomethane	cis-1,2-dichloroethene	1,1,1-trichloroethane
1,3-butadiene	1,2-dichloropropane	1,1,2-trichloroethane
carbon tetrachloride	cis-1,3-dichloropropene	trichloroethene
chlorobenzene	trans-1,3-dichloropropene	trichlorofluoromethane
chloroform	dichlorotetrafluoroethane	1,1,2-trichlorotrifluoroethane
chloromethane	ethyl benzene	1,2,4-trimethylbenzene
3-chloropropene	4-ethyltoluene	1,3,5-trimethylbenzene
1,2-dibromoethane	ethyl chloride	vinyl chloride
m-dichlorobenzene	hexachloro-1,3-butadiene	m-xylene
o-dichlorobenzene	methylene chloride	o-xylene
p-dichlorobenzene	styrene	p-xylene
dichlorodifluoromethane	1,1,2,2-tetrachloroethane	
1,1-dichloroethane	tetrachloroethylene	

Comment 8: Figure 3a and b should show control samples where the uncoated sensor has been first polluted with eg. N2 and then TO-14A + HCN is added, and no response is observed. This would indicate that the gatekeeper actually helps in preventing sensor fouling.

Response: We thank the Reviewer for pointing out how we can further clarify the mechanism of this sensor. Control experiments on an uncoated sensor can be found in **Extended Data Fig. S1**, where the effect of the zero-point drift in response to hexane infiltration can be observed. While reporting the change in *relative* intensity via the use of a reference peak helps to mitigate these effects when the two reflectance peaks exhibit comparable lateral shifts in wavelengths (as described in the previous manuscript on this sensing modality), at sufficiently high peak shifts the center of the signal peak will move outside of the absorbance peak of the CN-bound Cbi dye (**Extended Data Fig. S3**), causing the reflectance signal of the signal peak to be anomalously high, creating false negative results. This effect is precisely what is being observed in **Figure 3c** on the far-right, where a positive change in Response is observed after lattice opening. This can also be observed for the Co-free (i.e. permanently open) gatekeeper (**Figure 4b** and **Extended Data Figure S7**) as well as (to a much lesser extent) the non-switchable (i.e. permanently closed) ^{C98}RhuA gatekeeper (**Figure 4b** and **Extended Data Fig. S9**). Consequently, we believe that the before and after response to hexane depicted in **Figure 3c** clearly indicates the effect on Response fidelity when exposed to a mixture of fouling agent and analyte. To make this clearer to the readers, we have elaborated on the discussion of **Figure 3c**'s and **Figure 4**'s results (see also the comments below) as:

Response: We thank the Reviewer for pointing out how we can further clarify the mechanism of this sensor. Control experiments on an uncoated sensor can be found in **Extended Data Fig. S1**, where the effect of the zero-point drift in response to hexane infiltration can be observed. While reporting the change in *relative* intensity via the use of a reference peak helps to mitigate these effects when the two reflectance peaks exhibit comparable lateral shifts in wavelengths (as described in the previous manuscript on this sensing modality), at sufficiently high peak shifts the center of the signal peak will move outside of the absorbance peak of the CN-bound Cbi dye (**Extended Data Fig. S3**), causing the reflectance signal of the signal peak to be anomalously high, creating false negative results. This effect is precisely what is being observed in **Figure 3c** on the far-right, where a positive change in Response is observed after lattice opening. This can also be observed for the Co-free (i.e. permanently open) gatekeeper (**Figure 4b** and **Extended Data Figure S7**) as well as (to a much lesser extent) the non-switchable (i.e. permanently closed) ^{C98}RhuA gatekeeper (**Figure 4b** and **Extended Data Fig. S9**). Consequently, we believe that the before and after response to hexane depicted in **Figure 3c** clearly indicates the effect on Response fidelity when exposed to a mixture of fouling agent and analyte. To make this clearer to the readers, we have elaborated on the discussion of **Figure 3c**'s and **Figure 4**'s results (see also the comments below) as:

*“As the absorbance peak of CN-bound Cbi is fixed (**Extended Data Fig. S3**), sufficiently large shifts in the centers of the reflectance peaks due to infiltration of fouling agents can result in artificially high R_t/R_0 values even in the presence of HCN. The consequence can be observed in **Figure 3c** after HCN exposure, where subsequent hexane infiltration created a perceived change in the opposite direction of the signal from the HCN analyte; thus, it generated a false negative response in the pSi sensor when the gatekeeper was open, and no response when the gatekeeper was closed (**Fig. 3c**).”*

“Response of the samples to a flowing stream of n-hexane-saturated air (50% RH) showed that the permanently ajar “Co²⁺-free Gatekeeper” control was permeable to n-hexane, generating a positive zero-point drift in response to hexane infiltration in the pSi nanostructure that differed significantly from the other samples, reaching a positive drift of >0.05 from the starting point. As noted above, the positive drift in R_t/R_0 response following interferent infiltration can produce false negative signals in this sensor, highlighting the need for selective permeability.”

Comment 9: Related to Figure 4: How were the different protein films characterized and controlled to be similar / comparative. I would assume that e.g. coating thickness plays a major role.

Response: We took care to ensure the processing conditions were identical for each sensor type prepared. Visual inspection and optical reflectance measurements were performed to ensure consistency from sample to sample. All depositions of RhuA samples were performed with the same concentration of protein and Co (prepared from the same original stock solutions), ensuring consistent quantities of gatekeeper added to identically produced pSi chips. This is expected to result in fully comparable devices. Preparation details, including concentrations, are all provided in the methods in the section “*Synthesis, preparation, and deposition of gatekeeper crystals onto porous silicon (pSi) photonic crystals*”.

Comment 10: Figure 4b data lacks quantitative discussion. Are the observed differences statistically significant?

Response: We thank the reviewer for this observation. Figure 4b was discussed but was lacking specific figure callouts, which we have now added. We have modified the figure caption to define the measurements as: “Data points represent averages of three independent measurements on independent samples. Error bars are one standard deviation from the mean.”.

With this additional context, we also elaborated further in the subsection “*Validation of the Gatekeeper Function*” about these results: “The response of each of these controls, relative to the full gatekeeper sensor assembly, to HCN and hexanes is shown in **Figure 4b**. Response of the samples to a flowing stream of n-hexane-saturated air (50% RH) showed that the permanently ajar “Co²⁺-free Gatekeeper” control was permeable to n-hexane, generating a positive zero-point drift in response to hexane infiltration in the pSi nanostructure that differed significantly from the other samples, reaching a positive drift of >0.05 from the starting point. As noted above, the positive drift in R_t/R_0 Response following interferent infiltration can produce false negative signals in this sensor, highlighting the need for selective permeability. In contrast, the “¹CEERhuA Gatekeeper”, the “Disassembled Gatekeeper” and the “Non-Responsive ^{C98}RhuA Gatekeeper” lattices were all impermeable and did not respond to n-hexane, with triplicate measurements within error of each other and remaining close to the initial zero-point of 1.00 (**Figure 4b**). Consistent with the hexane response of each sample, exposure of the sensors to HCN triggered a strong response in the functional sensor (“¹CEERhuA Gatekeeper”) and in the always-open “Co²⁺-free Gatekeeper”, and no response in the “Disassembled Gatekeeper” and the “Non-Responsive ^{C98}RhuA Gatekeeper” (**Figure 4b**). The HCN response was tightly distributed around the mean for all except the “Co-Free Gatekeeper”, which exhibits large standard deviations in the Response curve, though it is distinct from the impermeable controls. It is possible that the hypothesized role of Co²⁺ in mediating interlayer stacking (**Figure 1c**) is important for a coherent gatekeeper response on the surface, and its absence leads to less-consistent deposition that could account for the wider variance of this specific control.

Of the above variants of the sensor, only the fully functional “¹CEERhuA Gatekeeper” sensor (as described in **Figure 1** and characterized in **Figure 3**) exhibited both HCN responsiveness and minimal zero-point drift (prior to HCN exposure). Crucially, the lack of response for the “^{C98}RhuA gatekeeper” demonstrates both that the presence of Co²⁺ alone is insufficient to produce a false-positive signal, and that conformational responsiveness of the crystal pores to metal binding is a requirement for selective permeability. Relatedly, the fact that the individual proteins can only occlude the surface but not selectively open in response to HCN (with all other aspects of the system remaining the same; **Fig. 4** and **Extended Data Fig. S8**) indicates that crystallinity is essential for the gatekeeper’s function.”

REVIEWER COMMENTS

Reviewer #1 (Remarks to the Author):

All my comments have been adequately addressed, therefore the manuscript can be accepted in its current form.

Reviewer #2 (Remarks to the Author):

The authors have made responses to the manuscript in line with the comments of the reviewers and the manuscript may be suitable for publication if the deficiencies noted by the other reviewers have been adequately addressed.

One minor point about the relative humidity: in work on sensing systems, we regularly experience R(H) over 90% so the authors claim of an upper limit of 80% being real-world is not accurate, in my opinion. Sensing responses can vary wildly under such extremity of condition especially where a water sensitive component of the system is present.

Reviewer #3 (Remarks to the Author):

The authors have answered to my questions and concerns almost fully by conducting extensive revisions and further experiments. My only remaining question concerns the original comment 10 about the significance of the observed responses. In Figure 4b, are the observed differences statistically significant e.g. after 600 s? This should be clearly indicated in the figure and discussion.

Responses to Reviewer Comments:

Reviewer 2:

“One minor point about the relative humidity: in work on sensing systems, we regularly experience R(H) over 90% so the authors claim of an upper limit of 80% being real-world is not accurate, in my opinion. Sensing responses can vary wildly under such extremity of condition especially where a water sensitive component of the system is present.”

While Reviewer 2 is correct in the literal sense (there are instances where RH is below 20% or above 80%), these are not common. Regardless, we have revised the corresponding sentence as follows:

“These results illustrate that the sensor did not give a false positive signal from the potential interferences, and it performed reliably over the range of humidity values between 20% and 80% RH. Humidity conditions above and below these ranges were not evaluated in this study.”

Reviewer 3:

“My only remaining question concerns the original comment 10 about the significance of the observed responses. In Figure 4b, are the observed differences statistically significant e.g. after 600 s? This should be clearly indicated in the figure and discussion.”

We have conducted additional statistical analyses (i.e., determination of P values) for the data described in Figure 10. These analyses are summarized in the revised caption for this figure, as well as in the new Extended Data Table 3.